# Justifying Born’s Rule *P_α_* = |Ψ*_α_*|^2^ Using Deterministic Chaos, Decoherence, and the de Broglie–Bohm Quantum Theory

**DOI:** 10.3390/e23111371

**Published:** 2021-10-20

**Authors:** Aurélien Drezet

**Affiliations:** Institut NEEL, CNRS and Université Grenoble Alpes, F-38000 Grenoble, France; aurelien.drezet@neel.cnrs.fr

**Keywords:** quantum probability, pilot-wave mechanics, entanglement, deterministic chaos

## Abstract

In this work, we derive Born’s rule from the pilot-wave theory of de Broglie and Bohm. Based on a toy model involving a particle coupled to an environment made of “qubits” (i.e., Bohmian pointers), we show that entanglement together with deterministic chaos leads to a fast relaxation from any statistical distribution ρ(x) of finding a particle at point *x* to the Born probability law ${\left|\Psi \left(x\right)\right|}^{2}$. Our model is discussed in the context of Boltzmann’s kinetic theory, and we demonstrate a kind of H theorem for the relaxation to the quantum equilibrium regime.

## 1. Introduction and Motivations

The work of Wojciech H. Zurek is universally recognized for its central importance in the field of quantum foundations; in particular, concerning decoherence and the understanding of the elusive border between the quantum and classical realms [1]. Zurek emphasized the role of pointer states and environment-induced superselection rules (einselections). In recent years, part of his work has gone beyond mere decoherence and averaging focused on quantum Darwinism and envariance. The main goal of quantum Darwinism is to emphasize the role of multiple copies of information records contained in the local quantum environment. Envariance aims is to justify the existence and form of quantum probabilities; i.e., deriving Born’s rule from specific quantum symmetries based on entanglement [2]. In recent important reviews of his work, Zurek stressed the importance of some of these concepts for discussing the measurement problem in relation with various interpretations of quantum mechanics [3,4]. Recent works showed, for instance, the importance of such envariance to the establishment of Born’s rule in the many-world and many-mind contexts [5,6]. While in his presentations, Zurek generally advocated a neutral position perhaps located between the Copenhagen and Everett interpretations, we believe his work on entanglement and decoherence could have a positive impact on other interpretations, such as the de Broglie–Bohm theory. We know that Zurek has always been careful concerning Bohmian mechanics (see for example his remarks in [7] p. 209) perhaps because of the strong ontological price one has to pay in order to assume a nonlocal quantum potential and surrealistic trajectories (present even if we include decoherence [3,8]). Moreover, the aim of this work is to discuss the pivotal role that quantum entanglement with an environment of “Bohmian pointers” could play in order to justify Born’s rule in the context of such a Bohmian interpretation. The goal is thus to suggest interesting and positive implications that decoherence could have on ontologies different from Everettian or consistent histories approaches. In this work, we were strongly inspired and motivated by the success of envariance for justifying quantum probabilities. Moreover, as mentioned above, Zurek’s envariance emphasizing the role of entanglement is more “interpretation independent”. Therefore, for comparison, we also include in the conclusion a short summary of Zurek’s proof for the Born rule and compare the result with ours.

The de Broglie–Bohm quantum theory (BBQT) introduced by de Broglie in 1927 [9,10,11] and further discussed by Bohm in 1952 [12,13], is now generally accepted as a satisfactory interpretation of quantum mechanics, at least for problems dealing with non-relativistic systems [14,15,16]. Within this regime, BBQT is a clean, deterministic formulation of quantum mechanics preserving the classical concepts of point-like particles moving continuously in space-time. This formulation is said to be empirically equivalent to the orthodox description axiomatized by the Copenhagen school, meaning that BBQT is able to justify and reproduce the probabilistic predictions made by the standard quantum measurement theory. More specifically, this implies recovering the famous Born rule, which connects the probability
(1)$$\begin{array}{c} P_{\alpha }={\left|\Psi_{\alpha }\right|}^{2} \end{array}$$
of observing an outcome α (associated with the quantum observable $\hat{A}$) to the amplitude $\Psi_{\alpha }$ in the quantum state expansion $|\Psi \rangle =\sum_{\alpha }\Psi_{\alpha }|\alpha \rangle$ (i.e., |α〉 is an eigenstate of $\hat{A}$ for the observable eigenvalue α).

This issue has been a recurrent subject of controversies since the early formulation of BBQT (see for example Pauli’s objection in [17,18]). It mainly arises because BBQT is a deterministic mechanics and therefore, like for classical statistical mechanics, probabilities in BBQT can only be introduced in relation with ignorance and uncertainty regarding the initial conditions of the particle motions. Moreover, after more than one and a half centuries of developments since the times of Maxwell and Boltzmann, it is well recognized that the physical and rigorous mathematical foundation of statistical mechanics is still debatable [19]. BBQT, which in some sense generalizes and extends Newtonian mechanics, clearly inherits these difficulties, constituting strong obstacles for defining a clean basis of its statistical formulation. This fact strongly contrasts with standard quantum mechanics, for which randomness has been axiomatized as genuine and inevitable from the beginning.

Over the years, several responses have been proposed by different proponents of BBQT to justify Born’s rule (for recent reviews, see [20,21,22]). Here, we would like to focus on the oldest approach, which goes back to the work of David Bohm on deterministic and molecular chaos. Indeed, in 1951–1952, Bohm already emphasized the fundamental role of the disorder and chaotic motion of particles for justifying Born’s rule [12,13]. In his early work, Bohm stressed that the complexity of the de Broglie–Bohm dynamics during interaction processes, such as quantum measurements, should drive the system to quantum equilibrium. In other words, during interactions with an environment such as a measurement apparatus, any initial probability distribution $\rho \left(X\right)\neq {\left|\Psi \left(X\right)\right|}^{2}$ for *N* particles in the configuration space (here $X=\left[x_{1},\ldots ,x_{M}\right]\in R^{3M}$ is a vector in the *N*-particles configuration space) should evolve in time to reach the quantum equilibrium limit $\rho \left(X\right)\to {\left|\Psi \left(X\right)\right|}^{2}$ corresponding to Born’s rule. In this approach, the relaxation process would be induced by both the high sensitivity to changes in the initial conditions of the particle motions (one typical signature of deterministic chaos) and by the molecular thermal chaos resulting from the macroscopic nature of the interacting environment (i.e., with $∼10^{23}$ degrees of freedom). Furthermore, in this strategy, Born’s rule $\rho \left(X\right)={\left|\Psi \left(X\right)\right|}^{2}$ should appear as an attractor similar to the microcanonical and canonical ensemble in thermodynamics. In 1953, Bohm developed an example model [23] (see [24] for a recent investigation of this idea) where a quantum system randomly submitted to several collisions with external particles constituting a bath was driven to quantum equilibrium $\rho \left(X\right)={\left|\Psi \left(X\right)\right|}^{2}$. In particular, during his analysis, Bohm sketched a quantum version of the famous Boltzmann *H*-theorem to prove the irreversible tendency to reach Born’s rule (for other clues that Bohm was already strongly fascinated by deterministic chaos in the 1950s, see [25] and the original 1951 manuscript written by Bohm in 1951 [26] and rediscovered recently).

However, in later works, especially in the work conducted with Vigier [27] and then subsequently Hiley [14], Bohm modified the original de Broglie–Bohm dynamics by introducing stochastic and fluctuating elements associated with a subquantum medium forcing the relaxation towards quantum equilibrium $\rho \left(X\right)\to {\left|\Psi \left(X\right)\right|}^{2}$. In this context, we mention that very important works have been done in recent years concerning “Stochastic Bohmian mechanics” based on the Schrödinger–Langevin framework, the Kostin equation and involving nonlinearities [28,29,30]. While this second semi-stochastic approach was motivated by general philosophical considerations [31], proponents of BBQT have felt divided concerning the need for such a modification of the original framework. In particular, starting in the 1990s, Valentini has developed an approach assuming the strict validity of BBQT as an underlying deterministic framework and introduced mixing and coarse-graining à la Tolman–Gibbs in the configuration space in order to derive a Bohmian “subquantum” version of the *H*-theorem [32,33]. However, we stress that the Tolman–Gibbs derivation [34] and therefore Valentini’s deduction can be criticized on many grounds (see for example [21] for a discussion). For instance, Prigogine already pointed out that the Tolman–Gibbs “proof” is a priori time-symmetric and cannot therefore be used to derive a relaxation. Furthermore, what the theorems show is that if we define a coarse-grained entropy $S{\left[\bar{\rho }\right]}_{t}$, we have necessarily (i.e., from the concavity of the entropy function) $S{\left[\bar{\rho }\right]}_{t}\geq S{\left[\rho \right]}_{t}=S{\left[\rho \right]}_{t=0}\equiv S{\left[\bar{\rho }\right]}_{t=0}$ (the second equality $S{\left[\rho \right]}_{t}=S{\left[\rho \right]}_{t=0}$ comes from unitarity and Liouville’s theorem, and the third one $S{\left[\rho \right]}_{t=0}\equiv S{\left[\bar{\rho }\right]}_{t=0}$ is an initial condition where the fine-grained and coarse-grained distributions are identical). However, this result cannot be used to directly prove the relation $S{\left[\bar{\rho }\right]}_{t+\delta }\geq S{\left[\bar{\rho }\right]}_{t}$ for δ≥0. In other words [21], one cannot show that the entropy is a monotonously growing function ultimately reaching quantum equilibrium (i.e., corresponding to the maximum of the entropy function [32]). Importantly, in his work on the “subquantum heat-death” (i.e., illustrated with many numerical calculations [35,36] often connected with cosmological studies [37,38]), Valentini and coworkers stressed the central role of deterministic chaos in the mixing processes, and this indeed leads to an increase of the entropy function in the examples considered. Moreover, deterministic chaos in BBQT is a research topic in itself (for a recent review, see [39,40]) and many authors (including Bohm [14] and Valentini [35,36]) have stressed the role of nodal-lines associated with phase-singularities of the wave-function for steering deterministic chaos in the BBQT [41,42,43]. However, it has also been pointed out [39,44] that this chaos is not generic enough to force the quantum relaxation $\rho \left(X\right)={\left|\Psi \left(X\right)\right|}^{2}$ for any arbitrary initial conditions $\rho \left(X\right)\neq {\left|\Psi \left(X\right)\right|}^{2}$ (a reversibility objection à la Kelvin–Loschmidt is already sufficient to see the impossibility of such an hypothetical deduction [21,45]). Therefore, this analysis ultimately shows that the H-theorem can only makes sense if we complete it with a discussion of the notion of typicality [45,46,47].

In the present work, we emphasize the role of an additional ingredient that (together with chaos and coarse graining) helps and steers the quantum dynamical relaxation $\rho \left(X\right)\to {\left|\Psi \left(X\right)\right|}^{2}$: quantum entanglement with the environment. The idea that quantum correlations must play a central role in BBQT for justifying Born’s rule is not new of course. Bohm already emphasized the role of entanglement in his work [13,14,23]. It has been shown that entanglement could lead to Born’s rule using ergodicity [48]. Moreover, in recent studies motivated by the Vigier–Bohm analysis, we developed a Fokker–Planck [22] and Langevin-like [49] description of relaxation to quantum equilibrium $\rho \left(X\right)={\left|\Psi \left(X\right)\right|}^{2}$ by coupling a small system *S* to a thermal bath or reservoir *T* inducing a Brownian motion on *S*. We showed that, under reasonable assumptions, we can justify a version of the *H*-theorem where quantum equilibrium appears as a natural attractor. Furthermore, at the end of [22], we sketched an even simpler strategy based on mixing together with entanglement and involving deterministic chaotic iterative maps. After the development of such an idea, it came to our attention that a similar strategy has been already developed in an elegant work by Philbin [50], and therefore we did not include too much detail concerning our model in [22]. Here, we present the missing part and provide a more complete and quantitative description of our scenario, which is presented as an illustration of a more general scheme. More precisely, we (i) analyze the chaotic character of the specific de Broglie–Bohm dynamics associated with our toy model, (ii) build a Boltzmann diffusion equation for the probability distribution and finally (iii) derive a simple *H*-theorem from which Born’s rule turns out to be an attractor. We emphasize that our work, like the one of Philbin, suggests interesting future developments for justifying Born’s rule and recovering standard quantum mechanics within BBQT.

## 2. The Status of Born’s Rule in the de Broglie–Bohm Theory

We start with the wave-function $\psi \left(x,t\right)=R\left(x,t\right)e^{iS(x,t)/\hbar }$ obeying Schrödinger’s equation
(2)$$\begin{array}{c} i\hbar \frac{\partial }{\partial t}\psi \left(x,t\right)=\frac{-\hbar^{2}\nabla^{2}}{2m}\psi \left(x,t\right)+V\left(x,t\right)\psi \left(x,t\right) \end{array}$$
for a single nonrelativistic particle with mass *m* in the external potentials V(x,t) (we limit the analysis to a single particle, but the situation is actually generic). BBQT leads to the first-order “guidance” law of motion
(3)$$\begin{array}{c} \frac{d}{dt}x^{\psi }\left(t\right)=v^{\psi }\left(x^{\psi }\left(t\right),t\right) \end{array}$$
where $v^{\psi }\left(x,t\right)=\frac{1}{m}\nabla S\left(x,t\right)$ defines an Eulerian velocity field and $x^{\psi }\left(t\right)$ is a de Broglie–Bohm particle trajectory. Furthermore, from Equation (Equation 2), we obtain the conservation rule:(4)$$\begin{array}{c} -\frac{\partial }{\partial t}R^{2}\left(x,t\right)=\nabla \cdot \left[R^{2}\left(x,t\right)v^{\psi }\left(x,t\right)\right] \end{array}$$
where we recognize $R^{2}\left(x,t\right)={\left|\psi \left(x,t\right)\right|}^{2}$ as the distribution which is usually interpreted as Born’s probability density. Now, in the abstract probability theory, we assign to every point x a density ρ(x,t) corresponding to a fictitious conservative fluid obeying the constraint
(5)$$\begin{array}{c} -\frac{\partial }{\partial t}\rho \left(x,t\right)=\nabla \cdot \left[\rho \left(x,t\right)v^{\psi }\left(x,t\right)\right]. \end{array}$$
Comparing with Equation (Equation 4), we deduce that the normalized distribution $f\left(x,t\right)=\frac{\rho (x,t)}{R^{2}\left(x,t\right)}$ satisfies the equation
(6)$$\begin{array}{c} \left[\frac{\partial }{\partial t}+v^{\psi }\left(x,t\right)\cdot \nabla \right]f\left(x,t\right):=\frac{d}{dt}f\left(x,t\right)=0. \end{array}$$
This actually means [23] that *f* is an integral of motion along any trajectory $x^{\psi }\left(t\right)$. In particular, if $f(x,t_{in})=1$ at a given time $t_{in}$ and for any point x, this holds true at any time *t*. Therefore, Born’s rule being valid at a given time will be preserved at any other time [11,12,23]. It is also important to see that the relation $\frac{d}{dt}f\left(x^{\psi }\left(t\right),t\right)=0$ plays the same role in BBQT for motions in the configuration space as Liouville’s theorem $\frac{d}{dt}\eta \left(q\left(t\right),p\left(t\right),t\right)=0$ in classical statistical mechanics (where η(q,p,t) is the probability density in phase space q,p). Therefore, with respect to the measure $d\Gamma ={\left|\psi \left(x,t\right)\right|}^{2}d^{3}x$ (which is preserved in time along trajectories since $\frac{d}{dt}d\Gamma_{t}=0$), the condition f=1 is equivalent to the postulate of equiprobability used in standard statistical mechanics for the microcanonical ensemble. Clearly, we see that the inherent difficulties existing in classical statistical mechanics to justify the microcanonical ensemble are transposed in BBQT to justify Born’s rule; i.e., f=1.

At that stage, the definition of the probability $\rho \left(x,t\right)d^{3}x$ of finding a particle in the infinitesimal volume $d^{3}x$ is rather formal and corresponds to a Bayesian–Laplacian interpretation where probabilities are introduced as a kind of measure of chance. Moreover, in BBQT, the actual and measurable density of particles must be defined using a “collective” or ensemble of *N*-independent systems prepared in similar quantum states $\psi (x_{i},t)$ with i=1,⋯,N. However, the concept of independency in quantum mechanics imposes the whole statistical ensemble with *N* particles to be described by the total factorized wave-function:(7)$$\begin{array}{c} \Psi_{N}\left(x_{1},\ldots ,x_{N},t\right)=\prod_{i=1}^{i=N}\psi \left(x_{i},t\right) \end{array}$$
as a solution of the equation
(8)$$\begin{array}{c} i\hbar \frac{\partial }{\partial t}\Psi_{N}=\left[\sum_{i=1}^{i=N}\frac{-\hbar^{2}\nabla_{i}^{2}}{2m}+V\left(x_{i},t\right)\right]\Psi_{N}. \end{array}$$
For this quantum state $\Psi_{N}$, BBQT allows us to build the velocity fields $\frac{d}{dt}x_{i}^{\psi }\left(t\right)=v^{\psi }\left(x_{i}^{\psi }\left(t\right),t\right)$, where $x_{i}^{\psi }\left(t\right):=x_{i}^{\Psi_{N}}\left(t\right)$ define the de Broglie–Bohm paths for the uncorrelated particles (i.e., guided by the individual and independent wave functions $\psi (x_{i},t)$ and Eulerian flows $v_{i}^{\Psi_{N}}\left(x_{1},\cdots ,x_{N},t\right)=v^{\psi }\left(x_{i},t\right)$). Within this framework, the actual density of particles P(r,t) at point r is given by
(9)$$\begin{array}{c} P\left(r,t\right)=\frac{1}{N}\sum_{k=1}^{k=N}\delta^{3}\left(r-x_{k}^{\psi }\left(t\right)\right) \end{array}$$
which clearly obeys the conservation rule
(10)$$\begin{array}{c} -\frac{\partial }{\partial t}P\left(x,t\right)=\nabla \cdot \left[P\left(x,t\right)v^{\psi }\left(x,t\right)\right]. \end{array}$$
Comparing with Equation (Equation 6), we see that if $\rho \left(x,t\right)={f\left(x,t\right)|\psi \left(x,t\right)|}^{2}$ plays the role of an abstract Laplacian probability, P(r,t) instead represents the frequentist statistical probability. Both concepts are connected by the weak law of large numbers (WLLN), which is demonstrated in the limit N→+∞ and leads to the equality ρ(x,t)=P(r,t); i.e.,
(11)$$\begin{array}{c} {f\left(r,t\right)|\psi \left(r,t\right)|}^{2}=\equiv \lim_{N\to +\infty }\frac{1}{N}\sum_{k=1}^{k=N}\delta^{3}\left(r-x_{k}^{\psi }\left(t\right)\right) \end{array}$$
where the equality must be understood in the sense of a “limit in probability” based on typicality and not as the more usual “point-wise limit”. We stress that the application of the WLLN already relies on the Laplacian notion of measure of chance since by definition in a multinomial Bernoulli process, the abstract probability density $\rho_{N}\left(x_{1},\cdots ,x_{N},t\right)=\prod_{i=1}^{i=N}\rho \left(x_{i},t\right)$ is used for weighting an infinitesimal volume of the *N*-particle configuration space $d\tau_{N}:=\prod_{i=1}^{i=N}d^{3}x_{i}$. It can be shown that in the limit N→+∞ with the use of this measure $\rho_{N}d\tau_{N}$, almost all possible configurations $x_{1}^{\psi }\left(t\right),\cdots ,x_{N}^{\psi }\left(t\right)$ obey the generalized Born’s rule $P\left(r,t\right)=\rho \left(x,t\right)={f\left(x,t\right)|\psi \left(x,t\right)|}^{2}$ (the fluctuation varying as $\frac{1}{\sqrt{N}}$). It is in that sense that Equation (Equation 11) is said to be typical, where typical means valid for “overwhelmingly many” cases; i.e., almost all states in the whole configuration space weighted by $\rho_{N}d\tau_{N}$. The application of the law of large numbers to BBQT is well known and well established [33,46,47] but has been the subject of intense controversies [45,46,51,52]. Issues concern (1) the interpretation of $\rho_{N}$ as a probability density—i.e., in relation with the notion of typicality—and (2) the choice of f=1 as natural and guided by the notion of equivariance [53]. To paraphrase David Wallace, the only thing the law of large numbers proves is that relative frequency tends to weight … with high weight [54]. However, there is a certain circularity in the reasoning that at least shows that the axiomatic nature of the probability calculus allows us to identify an abstract probability such as $\rho d^{3}x$ to a frequency of occurrence such as $Pd^{3}x$. However, the WLLN alone is unable to guide us in selecting a good measure for weighting typical configurations (the condition on equivariance [53] is only a convenient mathematical recipe based on elegant symmetries, not a physical consequence of a fundamental principle). Therefore, the value of the *f* function is unconstrained by the typicality reasoning without already assuming the result [51]. In other words, it is impossible to deduce Born’s rule directly from the WLLN.

However, it must be perfectly clear that our aim here is not to criticize the concept of typicality. Typicality, associated with the names of physicists such as Boltzmann or mathematicians such as Cournot and Borel, is, we think, at the core of any rigorous formulation of objective probability [55]. Our goal in the next section is to understand how natural and how stable the Born rule f=1 is. For this purpose, our method is to consider entanglement between an environment of pointers, already in quantum equilibrium, and a not yet equilibrated system driven by chaotic Bohmian dynamics to the quantum equilibrium regime.

## 3. A Deterministic and Chaotic Model for Recovering Born’s Rule within the de Broglie–Bohm Quantum Theory

### 3.1. The Basic Dynamics

As a consequence of the previous discussion, we now propose a simple toy model where the condition f=1 appears as an attractor; i.e, $f_{t}\to 1$ during a mixing process. We consider a single electron wave-packet impinging on a beam-splitter. To simplify the discussion, we consider an incident wave-train with one spatial dimension *x* characterized by the wave-function
(12)$$\begin{array}{c} \psi_{0}\left(x,t\right)≃\Phi_{0}\left(x-v_{x}t\right)e^{i(k_{x}x-\omega_{k}t)} \end{array}$$
where we have the dispersion relation $E_{k}:=\hbar \omega_{k}=\frac{\hbar^{2}k_{x}^{2}}{2m}$ and the (negative) group velocity components $v_{x}=\frac{\hbar k_{x}}{m}<0$ with $k_{x}=-\left|k_{x}\right|$. Furthermore, for mathematical consistency, we impose $\Phi_{0}≃const.=C$ in the spatial support region, where the wave-packet is not vanishing and the typical wavelength $\lambda =2\pi /|k_{x}|\ll L$, where *L* is a typical wave-packet spatial extension. If we assume Born’s rule, ${\left|C\right|}^{2}$ must be identified with a probability density, and by normalization this implies $C=1/\sqrt{L}$ (this point will be relevant only in Section 3). The beam-splitter is a rectangular potential barrier or well $V\left(x\right)=V_{0}$ with $V_{0}$ a constant in the region |x|<ϵ/2≪L and V(x)=0 otherwise. During the interaction with the beam-splitter, the whole wave-function approximately reads
(13)$$\begin{array}{c} \psi \left(x,t\right)≃\psi_{0}\left(x,t\right)+R_{k}\psi_{1}\left(x,t\right)\\ \mathrm{if}\;x>\epsilon /2\\ \psi \left(x,t\right)≃\Phi_{0}\left(-v_{x}t\right)\left[A_{k}e^{iq_{x}x}+B_{k}e^{-iq_{x}x}\right]e^{-i\omega_{k}t}\\ \mathrm{if}\;|x|<\epsilon /2\\ \psi \left(x,t\right)≃T_{k}\psi_{0}\left(x,t\right)\\ \mathrm{if}\;x<-\epsilon /2 \end{array}$$
where $\psi_{1}\left(x,t\right)=\Phi_{0}\left(x+v_{x}t\right)e^{-ik_{x}x}e^{-i\omega_{k}t}=\psi_{0}\left(-x,t\right)$ and $R_{k}$ (reflection amplitude), $T_{k}$(transmission amplitude) and $A_{k},B_{k}$ are Fabry–Perot coefficients computed in the limit where the wave-packet is infinitely spatially extended. We have(14)$$\begin{array}{c} T_{k}=\frac{4q_{x}k_{x}}{{\left(q_{x}+k_{x}\right)}^{2}}\frac{1}{e^{i(q_{x}-k_{x})\epsilon }-\frac{{\left(q_{x}-k_{x}\right)}^{2}}{{\left(q_{x}+k_{x}\right)}^{2}}e^{-i(k_{x}+q_{x})\epsilon }}\\ R_{k}=iT_{k}\frac{k_{x}^{2}-q_{x}^{2}}{2q_{x}k_{x}}\sin\left(q_{x}\epsilon \right)\\ A_{k}=R_{k}\left[\frac{q_{x}+k_{x}}{2q_{x}}e^{-i(q_{x}-k_{x})\epsilon /2}+\frac{q_{x}-k_{x}}{2q_{x}}e^{-i(q_{x}+k_{x})\epsilon /2}\right]\\ B_{k}=R_{k}\left[\frac{q_{x}-k_{x}}{2q_{x}}e^{i(q_{x}+k_{x})\epsilon /2}+\frac{q_{x}+k_{x}}{2q_{x}}e^{i(q_{x}-k_{x})\epsilon /2}\right] \end{array}$$
where $q_{x}$ is given by the dispersion relation $E_{k}:=\hbar \omega_{k}=\frac{\hbar^{2}q_{x}^{2}}{2m}+V_{0}$, i.e., $q_{x}^{2}-k_{x}^{2}=-2mV_{0}/\hbar$. As an illustration, we choose $\epsilon =\frac{1}{2}\frac{\lambda }{2\pi }$ and $q_{x}≃2.5k_{x}$ (i.e., $V_{0}<0$) which leads to $T_{k}≃\frac{1}{\sqrt{2}}e^{i0.267\pi }$ and $R_{k}=iT_{k}$ corresponding to a balanced 50/50 beam-splitter.

We consider the problem from the point of view of the scattering matrix theory. First, for negative time $t_{in}<0$ (with $|t_{in}\left|\gg L/\right|v_{x}|$), the incident wave-packet $\psi_{0}\left(x,t_{in}\right)$ given by Equation (Equation 12), which is coming from the x>0 region with a negative group velocity, is transformed for large positive times $t_{f}>0$ (with $|t_{f}\left|\gg L/\right|v_{x}|$) into the two non overlapping wave-packets: (15)$$\begin{array}{ccc} \psi \left(x,t_{f}\right)≃R_{k}\psi_{1}\left(x,t_{f}\right) & & \mathrm{if}\;x>0\\ \psi \left(x,t_{f}\right)≃T_{k}\psi_{0}\left(x,t_{f}\right) & & \mathrm{if}\;x<0. \end{array}$$
Since the wave packets are non-overlapping we write:(16)$$\begin{array}{c} \psi \left(x,t_{f}\right)≃R_{k}\psi_{1}\left(x,t_{f}\right)+T_{k}\psi_{0}\left(x,t_{f}\right). \end{array}$$
Of course, the situation is symmetric: if an incident wave-packet $\psi_{1}\left(x,t_{in}\right)$ comes from the x<0 region with a positive group velocity for $t_{in}<0$, we will finally obtain, i.e., for $t_{f}>0$,
(17)$$\begin{array}{c} \psi \left(x,t_{f}\right)≃T_{k}\psi_{1}\left(x,t_{f}\right)+R_{k}\psi_{0}\left(x,t_{f}\right). \end{array}$$
The general case can thus be treated by superposition: an arbitrary initial state $\psi \left(x,t_{in}\right)=a_{+}\psi_{0}\left(x,t_{in}\right)+a_{-}\psi_{1}\left(x,t_{in}\right)$ for negative times $t_{in}$ (with $|t_{in}\left|\gg L/\right|v_{x}|$) will evolve into
(18)$$\begin{array}{c} \psi \left(x,t_{f}\right)≃\left(a_{+}R_{k}+a_{-}T_{k}\right)\psi_{1}\left(x,t_{f}\right)\\ +\left(a_{+}T_{k}+a_{-}R_{k}\right)\psi_{0}\left(x,t_{f}\right) \end{array}$$
for positive times $t_{f}$ (with $|t_{f}\left|\gg L/\right|v_{x}|$). Writing $a_{+}^{\prime }=a_{+}R_{k}+a_{-}T_{k}$ and $a_{-}^{\prime }=a_{+}T_{k}+a_{-}R_{k}$ as the different mode amplitudes, we define a 2 × 2 unitary transformation
(19)$$\begin{array}{c} \left(\begin{array}{c} a_{+}^{\prime }\\ a_{-}^{\prime } \end{array}\right)=\left(\begin{array}{cc} R_{k} & T_{k}\\ T_{k} & R_{k} \end{array}\right)\left(\begin{array}{c} a_{+}\\ a_{-} \end{array}\right)\\ =\frac{e^{i0.267\pi }}{\sqrt{2}}\left(\begin{array}{cc} i & 1\\ 1 & i \end{array}\right)\left(\begin{array}{c} a_{+}\\ a_{-} \end{array}\right). \end{array}$$

Moreover, consider now the point of view of BBQT. Following this theory, the dynamics of the material point are obtained by the integration of the guidance equation
(20)ddtxψ(t)=vψ(xψ(t),t)=ℏmIm[∂∂xψ(x,t)|x=xψ(t)]
that can easily be computed numerically. We illustrate in Figure 1 the interaction with the 50/50 beam-splitter characterized by Equation (Equation 19) of a rectangular wave-packet (i.e., $\Phi_{0}\left(x\right)=C$ if |x|<L/2, where *L* is the width of the wave-packet) incident from the x>0 region (i.e., $a_{+}=1,a_{-}=0$). As a remarkable feature, we can see the Wiener fringes [11] existing in the vicinity of the beam-splitter and that strongly alter the de Broglie–Bohm trajectories. What is also immediately visible is that the de Broglie–Bohm trajectories $x^{\psi }\left(t\right)$ never cross each other. This is a general property of the first-order dynamics [14,15], which play a central role in our analysis.

An interesting feature of this example concerns the density of “probability” ${\left|\psi \left(x,t\right)\right|}^{2}$. Indeed, consider a time $t_{in}$ in the remote past before the wave-packet from the positive region (i.e., like in Figure 1) interacts with the potential well. At that time, the center of the wave-packet is located at $x_{in}=v_{x}t_{in}>0$. However, since trajectories cannot cross each other, we know that the ensemble $\gamma_{+}\left(t_{in}\right)$ of all possible particle positions at time $t_{in}$—i.e., $x^{\psi }\left(t_{in}\right)\in \left[x_{in}-\frac{L}{2},x_{in}+\frac{L}{2}\right]$—is divided into two parts. In the first part, $\gamma_{+}^{\left(+\right)}\left(t_{in}\right)$—i.e., $x^{\psi }\left(t_{in}\right)\in \left[x_{in}+H,x_{in}+\frac{L}{2}\right]$ with $|H|<\frac{L}{2}$—all particles evolve in the future (i.e., at time $t_{f}$) into the $\psi_{1}\left(x,t_{f}\right)$ reflected wave-packet (corresponding to the support $\gamma_{+}\left(t_{f}\right)$, i.e., $x^{\psi }\left(t_{f}\right)\in \left[x_{f}-\frac{L}{2},x_{f}-\frac{L}{2}\right]$ with $x_{f}=-v_{x}t_{f}>0$). In the second part $\gamma_{+}^{\left(-\right)}\left(t_{in}\right)$—i.e., $x^{\psi }\left(t_{in}\right)\in \left[x_{in}-\frac{L}{2},x_{in}+H\right]$—all the particles necessarily end their journey in the $\psi_{0}\left(x,t_{f}\right)$ transmitted wave-packet (corresponding to the support $\gamma_{-}\left(t_{f}\right)$, i.e., $x^{\psi }\left(t_{f}\right)\in \left[-x_{f}-\frac{L}{2},-x_{f}-\frac{L}{2}\right]$). Now, remember that from the de Broglie–Bohm–Liouville theorem, the measure $d\Gamma \left(x,t\right)={\left|\psi \left(x,t\right)\right|}^{2}dx$ is preserved in time; i.e., $\frac{d}{dt}d\Gamma_{t}=0$. Therefore, the measure
(21)$$\begin{array}{c} \Gamma_{+}\left(t_{f}\right)=\int_{\gamma_{+}}{\left|\psi \left(x,t\right)\right|}^{2}dx=LC^{2}/2 \end{array}$$
associated with the reflected wave necessarily equals the measure associated with the segment $\gamma_{+}^{\left(+\right)}\left(t_{in}\right)$; i.e.,
(22)$$\begin{array}{c} \Gamma_{+}^{\left(+\right)}\left(t_{in}\right)=\left(L/2-H\right)C^{2}=\Gamma_{+}\left(t_{f}\right). \end{array}$$
This leads to H=0, which in turn means that $\gamma_{+}^{\left(+\right)}\left(t_{in}\right)$ corresponds to $x^{\psi }\left(t_{in}\right)\in \left[x_{in},x_{in}+\frac{L}{2}\right]$ and $\gamma_{+}^{\left(-\right)}\left(t_{in}\right)$ to $x^{\psi }\left(t_{in}\right)\in \left[x_{in}-\frac{L}{2},x_{in}\right]$. This result is actually general and holds for any symmetric wave-packet $\Phi_{0}\left(x\right)=\Phi_{0}\left(-x\right)$ if we can neglect the overlap between $\Phi_{0}\left(x-v_{x}t_{f}\right)$ and $\Phi_{0}\left(x+v_{x}t_{f}\right)$).

Moreover, for the rectangular wave-packet, we deduce from the de Broglie–Bohm–Liouville theorem $\frac{d}{dt}d\Gamma_{t}=0$ that any infinitesimal-length element $\delta x^{\psi }\left(t_{in}\right)$ surrounding a point $x^{\psi }\left(t_{in}\right)$ in $\gamma_{+}\left(t_{in}\right)$ evolves to the infinitesimal length $\delta x^{\psi }\left(t_{f}\right)=2\delta x^{\psi }\left(t_{in}\right)$ surrounding the point $x^{\psi }\left(t_{f}\right)$ located in $\gamma_{\pm }\left(t_{f}\right)$. This property can be used to define a simple mapping between the initial coordinates $x^{\psi }\left(t_{in}\right)\in \gamma_{+}\left(t_{in}\right)$ and the final outcome $x^{\psi }\left(t_{f}\right)\in \gamma_{+}\left(t_{f}\right)\cup \gamma_{-}\left(t_{f}\right)$. It is simpler to introduce the normalized variables: (23)$$\begin{array}{ccc} y\left(t_{in}\right)=\frac{x^{\psi }\left(t_{in}\right)-x_{in}}{2L}+\frac{3}{4}\in \left[\frac{1}{2},1\right] & & \mathrm{if}\;x^{\psi }\left(t_{\mathrm{in}}\right)\in \gamma_{+}\left(t_{\mathrm{in}}\right)\\ y\left(t_{f}\right)=\frac{x^{\psi }\left(t_{f}\right)-x_{f}}{2L}+\frac{3}{4}\in \left[\frac{1}{2},1\right] & & \mathrm{if}\;x^{\psi }\left(t_{f}\right)\in \gamma_{+}\left(t_{f}\right)\\ y\left(t_{f}\right)=\frac{x^{\psi }\left(t_{f}\right)+x_{f}}{2L}+\frac{1}{4}\in \left[0,\frac{1}{2}\right] & & \mathrm{if}\;x^{\psi }\left(t_{f}\right)\in \gamma_{-}\left(t_{f}\right). \end{array}$$
The mapping between the two new ensembles (which we will continue to name $\gamma_{+}\left(t_{in}\right)$ and $\gamma_{+}\left(t_{f}\right)\cup \gamma_{-}\left(t_{f}\right)$) is thus simply written as
(24)$$\begin{array}{c} y\left(t_{f}\right)=2y\left(t_{in}\right)-1. \end{array}$$
The result of this mapping is illustrated using the *x* coordinates in Figure 2a or the *y* coordinates in Figure 2b. In particular, it is visible that the correspondence $y\left(t_{f}\right)=F\left(y\left(t_{in}\right)\right)$ is not always univocally defined. This occurs at $x^{\psi }\left(t_{in}\right)=x_{in}$ (i.e., $y\left(t_{in}\right)=\frac{3}{4}$), which evolves either as $x^{\psi }\left(t_{f}\right)=x_{f}-L/2\in \gamma_{+}\left(t_{f}\right)$ or $x^{\psi }\left(t_{f}\right)=-x_{f}+L/2\in \gamma_{-}\left(t_{f}\right)$ corresponding to the single value $y\left(t_{f}\right)=\frac{1}{2}$. Physically, as shown in Figure 2a, this means that a point located at the center of the wave-packet $\psi_{0}\left(x,t_{in}\right)$ is unable to decide whether it should move into the reflected or transmitted wave-packets: this is a point of instability. This apparently violates the univocity of the de Broglie–Bohm dynamics in Equation (Equation 20), which imposes that at a given point—i.e., $x^{\psi }\left(t_{in}\right)=x_{in}$—one and only one trajectory is defined. However, we stress that this pathology is actually a consequence of the oversimplification of our model consisting in assuming an idealized rectangular wave packet $\Phi_{0}\left(x\right)=C$ if |x|<L/2 with abrupt boundaries at |x|=L/2. In a real experiment with a Gaussian wave-packet, the point $x^{\psi }\left(t_{in}\right)=x_{in}$ would be mapped at the internal periphery of the two wave-packets constituting $\psi (x,t_{f})$ (this would correspond to the points $x^{\psi }\left(t_{f}\right)=\pm \epsilon /2∼0$ where the beam splitter is located). In this regime, our assumption of a finite support for $\Phi_{0}\left(x\right)$ is no longer acceptable.

The previous analysis was limited to the case of the wave-packet $\psi_{0}\left(x,t_{in}\right)$ coming from the x>0 region. However, in the symmetric case of a wave-packet $\psi_{1}\left(x,t_{in}\right)$ coming from the x<0 region (i.e., $a_{+}=0,a_{-}=1$), the situation is very similar (as shown in Figure 2), with the only differences being that the $\gamma_{+}\left(t_{in}\right)$ space is changed into $\gamma_{-}\left(t_{in}\right)$, i.e., $x^{\psi }\left(t_{in}\right)\in \left[-x_{in}-\frac{L}{2},-x_{in}+\frac{L}{2}\right]$ and the roles of $\gamma_{+}\left(t_{f}\right)$ and $\gamma_{-}\left(t_{f}\right)$ (the previous definitions are let unchanged) are now permuted (i.e., $\gamma_{+}\left(t_{f}\right)$ is now associated with the transmitted wave-packet and $\gamma_{-}\left(t_{f}\right)$ with the reflected one). From the point of view of BBQT, the trajectories of Figure 1b are obtained by a mirror symmetry x→−x from Figure 1a. The new mapping $x^{\psi }\left(t_{in}\right)\to x^{\psi }\left(t_{f}\right)$ is now well described by the variable transformation: (25)$$\begin{array}{ccc} y\left(t_{in}\right)=\frac{x^{\psi }\left(t_{in}\right)+x_{in}}{2L}+\frac{1}{4}\in \left[0,\frac{1}{2}\right] & & \mathrm{if}\;x^{\psi }\left(t_{\mathrm{in}}\right)\in \gamma_{-}\left(t_{\mathrm{in}}\right)\\ y\left(t_{f}\right)=\frac{x^{\psi }\left(t_{f}\right)-x_{f}}{2L}+\frac{3}{4}\in \left[\frac{1}{2},1\right] & & \mathrm{if}\;x^{\psi }\left(t_{f}\right)\in \gamma_{+}\left(t_{f}\right)\\ y\left(t_{f}\right)=\frac{x^{\psi }\left(t_{f}\right)+x_{f}}{2L}+\frac{1}{4}\in \left[0,\frac{1}{2}\right] & & \mathrm{if}\;x^{\psi }\left(t_{f}\right)\in \gamma_{-}\left(t_{f}\right). \end{array}$$
which lets the definition of $y(t_{f})$ unchanged with respect to Equation (Equation 24). The mapping between the two ensembles $\gamma_{-}\left(t_{in}\right)$ and $\gamma_{+}\left(t_{f}\right)\cup \gamma_{-}\left(t_{f}\right)$ is now written as
(26)$$\begin{array}{c} y\left(t_{f}\right)=2y\left(t_{in}\right) \end{array}$$
which is very similar to Equation (Equation 24).

### 3.2. Entanglement and Bernoulli’s Shift

If we regroup Equations (Equation 24) and (Equation 26) together with Equations (Equation 24) and (Equation 26), we are tempted to recognize the well known Bernoulli map:(27)$$\begin{array}{cc} y\left(t_{f}\right)=2y\left(t_{in}\right) & \mathrm{mod}(1), \end{array}$$
which actually means
(28)$$\begin{array}{cc} y\left(t_{f}\right)=2y\left(t_{in}\right)-1 & \mathrm{if}\;y\left(t_{\mathrm{in}}\right)>\frac{1}{2}\\ y\left(t_{f}\right)=2y\left(t_{in}\right) & \mathrm{if}\;y\left(t_{\mathrm{in}}\right)<\frac{1}{2} \end{array}$$
for $y(t_{f})$ and $y\left(t_{in}\right)\in \left[0,1\right]$. This would physically correspond to a mapping $\gamma_{+}\left(t_{in}\right)\cup \gamma_{-}\left(t_{in}\right)\to \gamma_{+}\left(t_{f}\right)\cup \gamma_{-}\left(t_{f}\right)$. In classical physics, such a mapping would be unproblematic since the two dynamics given by Equations (Equation 24) and (Equation 26) could be superposed without interference. However, in quantum mechanics, and specially in BBQT, the dynamics is contextually guided by the whole wave-function ψ(x,t) and a general superposition of states $\psi \left(x,t_{in}\right)=a_{+}\psi_{0}\left(x,t_{in}\right)+a_{-}\psi_{1}\left(x,t_{in}\right)$ evolves at $t_{f}$ to the state $\psi (x,t_{in})$ given by Equation (Equation 18). Consider for example with Equation (Equation 19) the unitary evolution
(29)$$\begin{array}{c} \frac{\psi_{0}\left(x,t_{in}\right)+i\psi_{1}\left(x,t_{in}\right)}{\sqrt{2}}\to ie^{i0.267\pi }\psi_{1}\left(x,t_{f}\right). \end{array}$$

From the point of view of BBQT (as illustrated in Figure 3), we have a mapping $\gamma_{+}\left(t_{in}\right)\cup \gamma_{-}\left(t_{in}\right)\to \gamma_{+}\left(t_{f}\right)$ which has nothing to do with either Equations (Equation 24) and (Equation 26) or even Equation (Equation 27). More precisely, the mapping associated with Equation (Equation 29) reads
(30)$$\begin{array}{c} y\left(t_{f}\right)=\frac{y(t_{in})}{2}+\frac{1}{2} \end{array}$$
Therefore, the high contextuality of the BBQT leads (in agreement with wave–particle duality) to new features induced by the coherence of the different branches of the input wave-function.

In order to make sense of the Bernoulli shift in Equation (Equation 27) in a simple way, we modify the properties of our beam-splitter by adding phase plates in the input and output channels. From here on, we consider instead of Equation (Equation 19) the unitary relation
(31)$$\begin{array}{c} \left(\begin{array}{c} a_{+}^{\prime }\\ a_{-}^{\prime } \end{array}\right)=\frac{1}{\sqrt{2}}\left(\begin{array}{cc} 1 & 1\\ 1 & -1 \end{array}\right)\left(\begin{array}{c} a_{+}\\ a_{-} \end{array}\right). \end{array}$$

Furthermore, in order to break the coherence between the two input waves $\psi_{0}\left(x,t_{in}\right)$ and $\psi_{1}\left(x,t_{in}\right)$, we introduce entanglement with an external pointer qubit before entering the beam splitter. The pointer must represent unambiguous “which-path” information concerning the moving particle in the context of BBQT. We represent the initial state of the pointer by a wave-function $\varphi_{in}^{1}\left(Z_{1}\right)$ associated with the coordinate $Z_{1}$ of the pointer (we assume $\int dZ_{1}{\left|\varphi_{in}^{1}\left(Z_{1}\right)\right|}^{2}=1$). The interaction leading to entanglement works in the following way: starting with an arbitrary state such as $A\psi_{0}\left(x,t_{0}\right)+B\psi_{1}\left(x,t_{0}\right)$ at time $t_{0}$ and a fixed initial pointer state $\varphi_{in}^{1}\left(Z_{1}\right)$, we obtain
(32)$$\begin{array}{c} (A\psi_{0}\left(x,t_{0}\right)+B\psi_{1}\left(x,t_{0}\right)\varphi_{in}^{1}\left(Z_{1}\right)\to A\psi_{0}\left(x,t_{0}\right)\varphi_{↑}^{1}\left(Z_{1}\right)+B\psi_{1}\left(x,t_{0}\right)\varphi_{↓}^{1}\left(Z_{1}\right). \end{array}$$
Here, we assume $\int dZ_{1}|\varphi_{↑}^{1}\left(Z_{1}\right){|}^{2}=\int dZ_{1}{\left|\varphi_{↓}^{1}\left(Z_{1}\right)\right|}^{2}=1$ and $\int dZ_{1}\varphi_{↑}^{1}\left(Z_{1}\right){\left(\varphi_{↓}^{1}\left(Z_{1}\right)\right)}^{*}=0$. Additionally, in order to simplify the analysis, we suppose the pointer–particle interaction to be quasi-instantaneous and act only at time $t≃t_{0}$. Moreover, in BBQT, the positions of the particle and pointer play a fundamental, ontic role. In order to have genuine Bohmian which-path information, we thus require that the two pointer wave-functions are well localized and are not overlapping; i.e., $\varphi_{↓}^{1}\left(Z_{1}\right)\varphi_{↑}^{1}\left(Z_{1}\right)=0$$\forall Z_{1}$.

We now consider the following sequences of processes, which are sketched in Figure 4. First, we prepare a non-entangled quantum system in the initial state $\psi_{0}\left(x,t_{0}^{\prime }\right)\varphi_{in}\left(Z\right)$ with $t_{0}^{\prime }\ll t_{0}$. Before interacting with the qubit, the particle wave-packet interacts with a first beam-splitter BS${}_{0}$, as in the previous subsection. Using Equations (Equation 31) and (Equation 32), this leads to
(33)$$\begin{array}{c} \psi_{0}\left(x,t_{0}^{\prime }\right)\varphi_{in}^{1}\left(Z_{1}\right)\to \frac{\psi_{1}\left(x,t_{0}\right)+\psi_{0}\left(x,t_{0}\right)}{\sqrt{2}}\varphi_{in}^{1}\left(Z_{1}\right)\to \frac{\psi_{1}\left(x,t_{0}\right)\varphi_{↑}^{1}\left(Z_{1}\right)+\psi_{0}\left(x,t_{0}\right)\varphi_{↓}^{1}\left(Z_{1}\right)}{\sqrt{2}}. \end{array}$$
In order to use a probabilistic interpretation—i.e., Born’s rule—we impose the normalization $C=1/\sqrt{L}$ associated with the wave-packet $\Phi_{0}$ (see Equation (Equation 12)). Second, as shown in Figure 4, the two wave-packets are moving in free space and interact with two mirrors which reflect the beams into the direction of a second beam-splitter BS${}_{1}$, where they cross (BS${}_{1}$ is the time translation of the same beam-splitter, but we continue to use this notation for simplicity). The main effect of the mirrors is to reverse the direction of propagation of $\psi_{0}\left(x,t_{0}\right)$ and $\psi_{1}\left(x,t_{0}\right)$—i.e., $\psi_{0}\left(x,t_{0}\right)\to -\psi_{1}\left(x,t_{1}^{\prime }+\frac{2D}{v_{x}}\right)e^{i\chi }$ and $\psi_{1}\left(x,t_{0}\right)\to -\psi_{0}\left(x,t_{1}^{\prime }+\frac{2D}{v_{x}}\right)e^{i\chi }$—with $t_{1}^{\prime }$ a time after the interaction and $\chi =\frac{2D}{v_{x}}\left(\omega_{k}-k_{x}v_{x}\right)$ a phase shift depending on the distance *D* between BS${}_{0}$ and any of the two mirrors ($-\frac{2D}{v_{x}}>0$ is the travel time taken by the center of the wave-packet for moving from BS${}_{0}$ to BS${}_{1}$). At a time $t_{1}^{\prime }$ before crossing BS${}_{1}$, the quantum state reads
(34)$$\begin{array}{c} -e^{i\chi }\frac{\psi_{0}\left(x,t_{1}^{\prime }+\frac{2D}{v_{x}}\right)\varphi_{↑}^{1}\left(Z_{1}\right)+\psi_{1}\left(x,t_{1}^{\prime }+\frac{2D}{v_{x}}\right)\varphi_{↓}^{1}\left(Z_{1}\right)}{\sqrt{2}}. \end{array}$$
At a time $t_{1}\gg -\frac{2D+L}{v_{x}}$ after the interaction with BS${}_{1}$ the quantum state reads (omitting the irrelevant phase factor)
(35)$$\begin{array}{c} \frac{\psi_{0}\left(x,t_{1}^{\prime }+\frac{2D}{v_{x}}\right)\varphi_{↑}^{1}\left(Z_{1}\right)+\psi_{1}\left(x,t_{1}^{\prime }+\frac{2D}{v_{x}}\right)\varphi_{↓}^{1}\left(Z_{1}\right)}{\sqrt{2}}\\ \to \frac{\psi_{1}\left(x,t_{1}+\frac{2D}{v_{x}}\right)\varphi_{\to }^{1}\left(Z_{1}\right)+\psi_{0}\left(x,t_{1}+\frac{2D}{v_{x}}\right)\varphi_{\leftarrow }^{1}\left(Z_{1}\right)}{\sqrt{2}} \end{array}$$
where $\varphi_{\to }^{1}=\frac{\varphi_{↑}^{1}+\varphi_{↓}^{1}}{\sqrt{2}}$ and $\varphi_{\leftarrow }^{1}=\frac{\varphi_{↑}^{1}-\varphi_{↓}^{1}}{\sqrt{2}}$ are two orthogonal eigenstates. Now, if we write this quantum state during the interaction with $BS_{1}$ as $\Psi \left(x,Z,t\right)=\psi_{↑}\left(x,t\right)\varphi_{↑}^{1}\left(Z_{1}\right)+\psi_{↓}\left(x,t\right)\varphi_{↓}^{1}\left(Z_{1}\right)$ we can define the Bohmian particle velocity $\frac{d}{dt}x^{\Psi }\left(t\right)=v\left(x,Z,t\right)$ as:(36)$$\begin{array}{c} \frac{d}{dt}x^{\Psi }\left(t\right)=\frac{v_{↑}\left(x,t\right)|\psi_{↑}\left(x,t\right)\varphi_{↑}^{1}\left(Z_{1}\right){|}^{2}+v_{↓}\left(x,t\right){\left|\psi_{↓}\left(x,t\right)\varphi_{↓}^{1}\left(Z_{1}\right)\right|}^{2}}{|\psi_{↑}\left(x,t\right)\varphi_{↑}^{1}\left(Z_{1}\right){|}^{2}+{\left|\psi_{↓}\left(x,t\right)\varphi_{↓}^{1}\left(Z_{1}\right)\right|}^{2}} \end{array}$$
where we introduced the two velocities $v_{↑/↓}\left(x,t\right)=\frac{1}{m}\partial_{x}S_{↑/↓}\left(x,t\right)$ associated with the two wave-functions $\psi_{↑/↓}\left(x,t\right)$. Equation (Equation 36) relies on the “which-path” constraint $\varphi_{↓}^{1}\left(Z_{1}\right)\varphi_{↑}^{1}\left(Z_{1}\right)=0$ and therefore we have here two different dynamics depending on the pointer position $Z_{1}$. If $Z_{1}$ lies in the support of $\varphi_{↑}^{1}\left(Z_{1}\right)$, we have the dynamics $\frac{d}{dt}x^{\Psi }\left(t\right)=v_{↑}\left(x,t\right)$ corresponding to Figure 1a, whereas if $Z_{1}$ lies in the support of $\varphi_{↓}^{1}\left(Z_{1}\right)$, we have the dynamics $\frac{d}{dt}x^{\Psi }\left(t\right)=v_{↓}\left(x,t\right)$ corresponding to Figure 1b.

The previous procedure for generating decohered Bohmian paths can be repeated iteratively at the times $t_{2}$, $t_{3}$, …after interaction with the beam-splitter BS${}_{2}$, BS${}_{3}$ …(see Figure 4). For this purpose, we consider at time $t_{1}$ entanglement with a an additional pointer initially in the state $\varphi_{in}^{2}(Z_{)}$, and we assume the transformation
(37)$$\begin{array}{c} \frac{\psi_{1}\left(x,t_{1}+\frac{2D}{v_{x}}\right)\varphi_{\to }^{1}\left(Z_{1}\right)+\psi_{0}\left(x,t_{1}+\frac{2D}{v_{x}}\right)\varphi_{\leftarrow }^{1}\left(Z_{1}\right)}{\sqrt{2}}\varphi_{in}^{2}\left(Z_{2}\right)\\ \to \frac{\psi_{1}\left(x,t_{1}+\frac{2D}{v_{x}}\right)\varphi_{\to }^{1}\left(Z_{1}\right)\varphi_{↑}^{2}\left(Z_{2}\right)+\psi_{0}\left(x,t_{1}+\frac{2D}{v_{x}}\right)\varphi_{\leftarrow }^{1}\left(Z_{1}\right)\varphi_{↓}^{2}\left(Z_{2}\right)}{\sqrt{2}}. \end{array}$$
The wave-packets propagate into the interferometer, and between times ${t^{\prime }}_{2}$ and $t_{2}$, we obtain
(38)$$\begin{array}{c} \frac{\psi_{0}\left(x,t_{2}^{\prime }+\frac{4D}{v_{x}}\right)\varphi_{\to }^{1}\left(Z_{1}\right)\varphi_{↑}^{2}\left(Z_{2}\right)+\psi_{1}\left(x,t_{2}^{\prime }+\frac{4D}{v_{x}}\right)\varphi_{\leftarrow }^{1}\left(Z_{1}\right)\varphi_{↓}^{2}\left(Z_{2}\right)}{\sqrt{2}}\\ \to \frac{\psi_{1}\left(x,t_{2}+\frac{4D}{v_{x}}\right)\varphi_{\to }^{12}\left(Z_{1},Z_{2}\right)+\psi_{0}\left(x,t_{2}+\frac{4D}{v_{x}}\right)\varphi_{\leftarrow }^{12}\left(Z_{1},Z_{2}\right)}{\sqrt{2}} \end{array}$$
with the orthonormal states $\varphi_{\to }^{12}=\frac{1}{\sqrt{2}}\left(\varphi_{\to }^{1}\varphi_{↑}^{2}+\varphi_{\leftarrow }^{1}\varphi_{↓}^{2}\right)$ and $\varphi_{\leftarrow }^{12}\left(Z_{1},Z_{2}\right)=\frac{1}{\sqrt{2}}\left(\varphi_{\to }^{1}\varphi_{↑}^{2}-\varphi_{\leftarrow }^{1}\varphi_{↓}^{2}\right)$.

This can be generalized at any time $t_{n}$ after interaction with BS${}_{n}$:(39)$$\begin{array}{c} \frac{\psi_{0}\left(x,t_{n}^{\prime }+\frac{2nD}{v_{x}}\right)\varphi_{\to }^{1,\cdots ,n-1}\left(Z_{1},\cdots ,Z_{n-1}\right)\varphi_{↑}^{n}\left(Z_{n}\right)+\psi_{1}\left(x,t_{n}^{\prime }+\frac{2nD}{v_{x}}\right)\varphi_{\leftarrow }^{1,\cdots ,n-1}\left(Z_{1},\cdots ,Z_{n-1}\right)\varphi_{↓}^{n}\left(Z_{n}\right)}{\sqrt{2}}\\ \to \frac{\psi_{1}\left(x,t_{n}+\frac{2nD}{v_{x}}\right)\varphi_{\to }^{1,\cdots ,n}\left(Z_{1},\cdots ,Z_{n}\right)+\psi_{0}\left(x,t_{n}+\frac{2nD}{v_{x}}\right)\varphi_{\leftarrow }^{1,\cdots ,n}\left(Z_{1},\cdots ,Z_{n}\right)}{\sqrt{2}}, \end{array}$$
with the orthonormal states $\varphi_{\to /\leftarrow }^{1,\cdots ,n}=\frac{1}{\sqrt{2}}\left(\varphi_{\to }^{1,\cdots ,n-1}\varphi_{↑}^{n}\pm \varphi_{\leftarrow }^{1,\ldots ,n-1}\varphi_{↓}^{n}\right)$. Like for the interaction at BS${}_{1}$ (between $t_{1}^{\prime }$ and $t_{1}$), we can define a Bohmian dynamical evolution similar to Equation (Equation 36) but based on the wave-function
(40)$$\begin{array}{c} \Psi \left(x,Z1,\cdots ,Z_{n},t\right)=\psi_{↑}\left(x,t\right)\varphi_{\to }^{1\cdots ,n-1}\left(Z_{1},\cdots ,Z_{n-1}\right)\varphi_{↑}^{n}\left(Z_{n}\right)\\ +\psi_{↓}\left(x,t\right)\varphi_{\leftarrow }^{1,\cdots ,n-1}\left(Z_{1},\cdots ,Z_{n-1}\right)\varphi_{↓}^{2}\left(Z_{n}\right). \end{array}$$
We obtain the velocity
(41)$$\begin{array}{c} \frac{d}{dt}x^{\Psi }\left(t\right)=\frac{v_{↑}\left(x,t\right){\left|\psi_{↑}\left(x,t\right)\varphi_{\to }^{1,\ldots ,n-1}\left(Z_{1},\ldots ,Z_{n-1}\right)\varphi_{↑}^{n}\left(Z_{n}\right)\right|}^{2}}{|\psi_{↑}\left(x,t\right)\varphi_{\to }^{1,\ldots ,n-1}\left(Z_{1},\ldots ,Z_{n-1}\right)\varphi_{↑}^{n}\left(Z_{n}\right){|}^{2}+{\left|\psi_{↓}\left(x,t\right)\varphi_{\leftarrow }^{1,\ldots ,n-1}\left(Z_{1},\ldots ,Z_{n-1}\right)\varphi_{↓}^{2}\left(Z_{n}\right)\right|}^{2}}\\ +\frac{v_{↓}\left(x,t\right){\left|\psi_{↓}\left(x,t\right)\varphi_{\leftarrow }^{1,\ldots ,n-1}\left(Z_{1},\ldots ,Z_{n-1}\right)\varphi_{↓}^{2}\left(Z_{n}\right)\right|}^{2}}{|\psi_{↑}\left(x,t\right)\varphi_{\to }^{1,\ldots ,n-1}\left(Z_{1},\ldots ,Z_{n-1}\right)\varphi_{↑}^{n}\left(Z_{n}\right){|}^{2}+{\left|\psi_{↓}\left(x,t\right)\varphi_{\leftarrow }^{1,\ldots ,n-1}\left(Z_{1},\ldots ,Z_{n-1}\right)\varphi_{↓}^{2}\left(Z_{n}\right)\right|}^{2}} \end{array}$$
which like Equation (Equation 36) reduces to one of the two dynamics (i) $\frac{d}{dt}x^{\Psi }\left(t\right)=v_{↑}\left(x,t\right)$ if $Z_{n}$ lies in the support of $\varphi_{↑}^{n}\left(Z_{n}\right)$ (i.e., corresponding to Figure 1a) or (ii) $\frac{d}{dt}x^{\Psi }\left(t\right)=v_{↓}\left(x,t\right)$ if $Z_{n}$ lies in the support of $\varphi_{↓}^{n}\left(Z_{n}\right)$ (i.e., corresponding to Figure 1b). The full history of the particle in the interferometer depends on the positions $Z_{1},\ldots ,Z_{n}$ taken by the various Bohmian pointers. In turn, this deterministic iterative process allows us to define a Bernoulli map for the evolution.

### 3.3. Mixing, Chaos and Relaxation to Quantum Equilibrium

The Bernoulli map is clearly defined from Equations (Equation 27) and (Equation 28) after introducing the variable y(t) replacing x(t). Between $t_{n}^{\prime }$ and $t_{n}$, this reads
(42)$$\begin{array}{cc} y\left(t_{n}\right)=2y\left(t_{n}^{\prime }\right) & \mathrm{mod}(1). \end{array}$$
Moreover, the $y(t_{n}^{\prime })$ coordinate at time $t_{n}^{\prime }$ is obviously equal to $y(t_{n-1})$ at time $t_{n-1}$ (see Figure 4), and therefore we have the map
(43)$$\begin{array}{cc} y\left(t_{n}\right)=2y\left(t_{n-1}\right) & \mathrm{mod}(1). \end{array}$$
This iterative Bernoulli map $y_{n}=F\left(y_{n-1}\right)$ is one of the simplest chaotic maps discussed in the literature [56,57]. In particular, its chaotic nature has been already studied in the context of BBQT [58,59] (for different purposes than those considered here), and an attempt to use it for deriving Born’s rule has been worked out [60] (without the entanglement used here and in [22,50]).

The chaotic nature of the map is easy to obtain; consider for example Figure 5.

In Figure 5a, we show a standard representation of the iterative function $y_{n}=F\left(y_{n-1}\right)$ for two paths initially starting at $y_{0}=0.22$ and $y_{0}=0.23$, and after a few iterations, the coordinates are apparently diverging in an unpredictable way. This is even more clear in the representation of Figure 5b, where two trajectories $y\left(t_{n}\right):=y_{n}$ are shown with $y_{0}=0.22$ and $y_{0}=0.220001$. Again, the motions become chaotic after a few iterations, and the trajectories are strongly diverging. Mathematically, any number *y* in the interval [0,1] is represented in binary notation as $0.u_{1}u_{2}\ldots u_{n}\ldots$, i.e., $y=\frac{u_{1}}{2}+\frac{u_{2}}{4}+\ldots +\frac{u_{n}}{2^{n}}+\ldots$ where $u_{n}=0$ or 1. The Bernoulli transformation $y^{\prime }=F\left(y\right)$ with $y^{\prime }=\frac{u_{1}^{\prime }}{2}+\frac{u_{2}^{\prime }}{4}+\ldots +\frac{u_{n}^{\prime }}{2^{n}}+\ldots$ corresponds to the shift $u_{n}^{\prime }=u_{n-1}$; i.e., to the binary number $0.u_{2}u_{3}\ldots u_{n-1}\ldots$. Iteratively, this generates chaos since if the $n^{th}$ term in $y=\frac{u_{1}}{2}+\frac{u_{2}}{4}+\ldots +\frac{u_{n}}{2^{n}}+\ldots$ is known with an uncertainty $\delta y_{i}=\frac{1}{2^{n}}$ after *n* iterations, this uncertainty will grow to $\delta y_{f}=1/2$. For example, if n=133 and $\delta y_{i}=2^{-133}≃10^{-40}$, we have after only 40 iterations completely lost any predictability in the dynamics (note that rational numbers are periodical in the binary representation and therefore the sequence will reappear periodically for rational numbers representing a null measure in the segment [0,1]). It can be shown that this feature leads to randomness in close analogy with ideal probabilistic coin tossing [61]. Therefore, any uncertainty will ultimately lead to chaos. The Lyapunov divergence of this Bernoulli map is readily obtained by considering as in Figure 5 two trajectories $y_{n}^{\left(A\right)}$ and $y_{n}^{\left(B\right)}=y_{n}^{\left(A\right)}+\delta y_{n}$ differing by a infinitesimal number such that
(44)$$\begin{array}{c} \delta y_{n}=2\delta y_{n-1}=2^{n}\delta y_{0}=e^{n\ln2}\delta y_{0} \end{array}$$
where the positive Lyapunov exponent ln2 characterizes the chaotic dynamics. If we introduce the time delay $\delta t=-2D/v_{x}>0$ and define the evolution time as $t_{n}=n\delta t$, we can rewrite the exponential divergence in Equation (Equation 44) as $e^{+t/\tau }$ where $\tau =\frac{\delta t}{\ln2}$ defines a Lyapunov time.

Most importantly, the Bernoulli shift allows us to define a mixing property for the probability distribution ρ(y). More precisely, we can consider at any time $t_{n}$ the probability density $\rho \left(x,t_{n}\right)=\int \ldots \int \rho \left(x,Z_{1},\ldots ,Z_{n},t_{n}\right)dZ_{1}\ldots dZ_{n}$, where according to BBQT we have $\rho \left(x,Z_{1},\ldots ,Z_{n},t_{n}\right)=f\left(x,Z_{1},\ldots ,Z_{n},t_{n}\right){\left|\Psi \left(x,Z_{1},\ldots Z_{n},t_{n}\right)\right|}^{2}$. In this framework, $\rho (x,t_{n})$ is a coarse-grained probability involving a form of classical ignorance. In the following, we suppose that the pointers are all in quantum equilibrium, and we have $f\left(x,Z_{1},\ldots ,Z_{n},t_{n}\right):=f\left(x,t_{n}\right)$ and $\rho \left(x,t_{n}\right)dx=f\left(x,t_{n}\right)d\Gamma \left(x,t_{n}\right)$ with $d\Gamma \left(x,t_{n}\right)=dx\int \ldots \int {\left|\Psi \left(x,Z_{1},\ldots Z_{n},t_{n}\right)\right|}^{2}dZ_{1}\ldots dZ_{n}$.

For the present purpose, a key result of deterministic maps such as $y_{n}=F\left(y_{n-1}\right)$ is the Perron–Frobenius theorem [56,57] allowing us to introduce the operator ${\hat{U}}_{PF}$; i.e., $\mu \left(y,t_{n+1}\right)={\hat{U}}_{PF}\mu \left(y,t_{n}\right)$ with the definition ρ(x,t)dx=μ(y,t)dy. For this, we use the property for a trajectory
(45)$$\begin{array}{c} \delta \left(y-y_{n+1}\right)=\delta \left(y-F\left(y_{n}\right)\right)=\int_{0}^{1}dY\delta \left(y-F\left(Y\right)\right)\delta \left(Y-y_{n}\right) \end{array}$$
and the fact that any density $\mu (w,t_{n})$ reads
(46)$$\begin{array}{c} \int_{0}^{1}dy\left(t_{n}\right)\mu \left(y\left(t_{n}\right),t_{n}\right)\delta \left(w-y\left(t_{n}\right)\right)=\int_{0}^{1}dy\left(t_{0}\right)\mu \left(y\left(t_{0}\right),t_{0}\right)\delta \left(w-y\left(t_{n}\right)\right) \end{array}$$
(where we used Liouville’s theorem $dy\left(t_{n}\right)\mu \left(y\left(t_{n}\right),t_{n}\right)=dy\left(t_{0}\right)\mu \left(y\left(t_{0}\right),t_{0}\right)$). Therefore, from Equation (Equation 45), we obtain
(47)$$\begin{array}{c} \mu \left(y,t_{n+1}\right)={\hat{U}}_{PF}\mu \left(y,t_{n}\right)=\int_{0}^{1}dY\delta \left(y-F\left(Y\right)\right)\mu \left(Y,t_{n}\right) \end{array}$$
which for the Bernoulli map means
(48)$$\begin{array}{c} \mu \left(y,t_{n+1}\right)={\hat{U}}_{PF}\mu \left(y,t_{n}\right)=\frac{1}{2}\left[\mu \left(\frac{y}{2},t_{n}\right)+\mu \left(\frac{y+1}{2},t_{n}\right)\right]. \end{array}$$
Moreover, for the present wave-function defined in term of the wave-packet $\Phi_{0}\left(x\right)$ which is constant in amplitude in its support, we can also write
(49)$$\begin{array}{c} \tilde{f}\left(y,t_{n+1}\right)={\hat{U}}_{PF}\tilde{f}\left(y,t_{n}\right)=\frac{1}{2}\left[\tilde{f}\left(\frac{y}{2},t_{n}\right)+\tilde{f}\left(\frac{y+1}{2},t_{n}\right)\right] \end{array}$$
with $f\left(x,t\right)=\tilde{f}\left(y,t\right)$ using the transformation x→y (see Equations (Equation 24) and (Equation 26)) by definition and where $\int_{\gamma_{+}\left(t_{n}\right)\cup \gamma_{-}\left(t_{n}\right)}dx\frac{{\left|C\right|}^{2}}{2}f\left(x,t_{n}\right)=\int_{0}^{1}dy\tilde{f}\left(y,t_{n}\right)=1$ involving the normalization $C=1/\sqrt{L}$. This iterative Perron–Frobenius relation admits Bernoulli polynomial eigenstates defined by $\frac{1}{2^{n}}B_{n}\left(y\right)={\hat{U}}_{PF}B_{n}\left(y\right)$ with $B_{0}\left(y\right)=1$, $B_{1}\left(y\right)=y-1/2$, $B_{2}\left(y\right)=y^{2}-y+1/6$, … [56].

It can be shown [56] that the $B_{m}\left(y\right)$ polynomials form a basis for the probability function $\tilde{f}\left(y,t\right)$, and therefore we write $\tilde{f}\left(y,t_{0}\right)=\sum_{m=0}^{m=+\infty }A_{m}B_{m}\left(y\right)$, which we obtain after *n* iterations of the ${\hat{U}}_{PF}–$operator: (50)$$\begin{array}{c} \tilde{f}\left(y,t_{n}\right)=\sum_{m=0}^{m=+\infty }A_{m}e^{-n\cdot m\ln2}B_{m}\left(y\right). \end{array}$$
In this formula, we have [56,62]
(51)$$\begin{array}{c} A_{m}=\int_{0}^{1}dy\tilde{f}\left(y,t_{0}\right){\tilde{B}}_{m}\left(y\right) \end{array}$$
where ${\tilde{B}}_{0}\left(y\right)=1$ and ${\tilde{B}}_{m}\left(y\right)=\lim_{\varepsilon \to 0^{+}}\frac{{\left(-1\right)}^{m-1}}{m!}\frac{d^{m-1}}{dy^{m-1}}\left[\delta \left(y-1+\varepsilon \right)-\delta \left(y-\varepsilon \right)\right]$ for m≥1. This leads to $A_{0}=\int_{0}^{1}\tilde{f}\left(y,t_{0}\right)dy$ and $A_{m}=\lim_{\varepsilon \to 0^{+}}\frac{1}{m!}\frac{d^{m-1}}{dy^{m-1}}\left[\tilde{f}\left(1-\varepsilon ,t_{0}\right)-\tilde{f}\left(0+\varepsilon ,t_{0}\right)\right]$. Equation (Equation 50) is important as it shows that in the limit n→+∞, we necessarily have $\tilde{f}\left(y,t_{n}\right)\to A_{0}B_{0}\left(y\right)=A_{0}$. Moreover, from the properties of the Bernoulli polynomials and the normalization of the probability density, we necessarily have $\int_{0}^{1}\tilde{f}\left(y,t\right)dy=A_{0}=1$ (with $\int_{0}^{1}dyB_{m}\left(y\right)=\delta_{0,m}$). Therefore, we deduce
(52)$$\begin{array}{c} \lim_{n\to +\infty }\tilde{f}\left(y,t_{n}\right)=\lim_{n\to +\infty }f\left(x,t_{n}\right)=1. \end{array}$$

This result says that quantum equilibrium, and therefore Born’s rule, is a statistical attractor in BBQT. Importantly, Equation (Equation 50) shows that each term in the sum is characterized by an exponential decay $e^{-mt_{n}/\tau }$, which is a signature of stability (negative Lyapunov exponent) whereas the trajectories (as we have shown in Equation (Equation 44)) have a positive Lyapunov exponent associated with dynamical instability and chaos. These two pictures are thus clearly complementary. This was already emphasized long ago by Prigogine in a different context [62,63]. As an illustration, we show in Figure 6 the transformation of an arbitrary (normalized) density $\tilde{f}\left(y,t_{0}\right)$: after only three applications of the Perron–Frobenius operator, the density is indistinguishable from the quantum equilibrium $\tilde{f}=f=1$, which acts as a very efficient attractor.

We emphasize that the iterative process sketched in Figure 4 and associated with states such as Equations (Equation 39) and (Equation 40) ultimately involves the two branches $\psi_{0}\left(x,t_{n}\right)$ and $\psi_{1}\left(x,t_{n}\right)$ entangled with an environment of Bohmian pointers characterized by $\varphi_{\to /\leftarrow }^{1,\ldots ,n}=\frac{1}{\sqrt{2}}\left(\varphi_{\to }^{1,\ldots ,n-1}\varphi_{↑}^{n}\pm \varphi_{\leftarrow }^{1,\ldots ,n-1}\varphi_{↓}^{n}\right)$. Moreover, because of the orthogonality of these pointer states, the two branches $\psi_{0}\left(x,t_{n}\right)$ and $\psi_{1}\left(x,t_{n}\right)$ cannot interfere: they are decohered. Still, in each of the two final wave-packets $\psi_{0}\left(x,t_{N}\right)$ and $\psi_{1}\left(x,t_{N}\right)$ (after a large number of iterations *N*), we have $f(x,t_{N})≃1$ with a high accuracy. Therefore, supposing that we now make a pinhole to select one of these two branches, we have prepared a quantum system satisfying Born’s rule $\rho \left(x,t\right)≃{\left|\psi \left(x,t\right)\right|}^{2}$. Fundamentally, this means that if an entangled system such as the system we discussed is post-selected by a filtering procedure, we can define subsystems for which Born’s rule is true and where quantum coherence is maintained (this is the case with our two wave-functions $\psi_{1}$ and $\psi_{0}$ taken separately). For example, the wave-function $\psi_{0}\left(x,t\right)$ can be collimated and sent into an interferometer in order to observe wave–particle duality. All systems following this guiding wave belong to a statistical ensemble of particles obeying Born’s rule f≃1. Therefore, all the predictions of standard quantum mechanics are reproduced with these systems.

Although the present model is rudimentary, it allows us to obtain precious information on relaxation to quantum equilibrium. Indeed, observe that in the continuous time approximation, we have $\tilde{f}\left(y,t\right)≃1+A_{1}e^{-t/\tau }B_{1}\left(y\right)$, which is a solution of the differential equation
(53)$$\begin{array}{c} \frac{\partial \tilde{f}\left(y,t\right)}{\partial t}=-\frac{\tilde{f}\left(y,t\right)-1}{\tau } \end{array}$$
This suggests a collision term in a Boltzmann-like equation and therefore an extension of our model by writing
(54)$$\begin{array}{c} \frac{df(x,t)}{dt}:=\partial_{t}f\left(x,t\right)+v_{\psi }\left(x,t\right)\partial_{x}f\left(x,t\right)=-\frac{f(x,t)-1}{\tau } \end{array}$$
or equivalently with $\rho \left(x,t\right)={f\left(x,t\right)|\psi \left(x,t\right)|}^{2}$ and $\partial_{t}{\left|\psi \left(x,t\right)\right|}^{2}+\partial_{x}(v_{\psi }{\left(x,t\right)|\psi \left(x,t\right)|}^{2})=0$:(55)$$\begin{array}{c} \partial_{t}\rho \left(x,t\right)+\partial_{x}\left(v_{\psi }\left(x,t\right)\rho \left(x,t\right)\right)=-\frac{\rho \left(x,t\right)-{\left|\psi \left(x,t\right)\right|}^{2}}{\tau }. \end{array}$$
With such dynamics (with an effective broken time symmetry), it is useful to introduce the Valentini entropy [32]:(56)$$\begin{array}{c} S_{t}:=-\int f\left(x,t\right)\ln\left(f(x,t)\right)d\Gamma \left(x,t\right) \end{array}$$
with $d\Gamma \left(x,t\right)={\left|\psi \left(x,t\right)\right|}^{2}$. From the previous equation, we deduce
(57)$$\begin{array}{c} \frac{d}{dt}S_{t}=-\int \frac{df_{t}}{dt}\left(1+\ln f_{t}\right)d\Gamma_{t}=\int \frac{\left(f_{t}-1\right)}{\tau }\left(1+\ln f_{t}\right)d\Gamma_{t}=\int \frac{\left(f_{t}-1\right)}{\tau }\ln f_{t}d\Gamma_{t}. \end{array}$$
This kinetic equation leads to a quantum version of the Boltzmann *H*-theorem, as can be shown easily: first, we have by definition $a\ln b+\frac{a}{b}-a\geq 0$ (with a,b>0) leading to $\left(f-1\right)\ln f+\frac{f-1}{f}-f-1\geq 0$ if f−1>0; i.e., we obtain $\left(f-1\right)\ln f\geq \frac{{\left(f-1\right)}^{2}}{f}$ if f−1>0. Moreover, lnf≤f−1, and thus if f−1<0, we have $\left(f-1\right)\ln f\geq {\left(f-1\right)}^{2}$. Now, separating the full Γ− space into two parts $\Gamma_{+}$ and $\Gamma_{-}$ where f−1≥0 and 1−f≥0, respectively, we have
(58)$$\begin{array}{c} \frac{d}{dt}S_{t}=\int \frac{\left(f_{t}-1\right)}{\tau }\ln f_{t}d\Gamma_{t}\geq \int_{\Gamma_{+}}\frac{{\left(f_{t}-1\right)}^{2}}{f_{t}\tau }d\Gamma_{t}+\int_{\Gamma_{-}}\frac{{\left(f_{t}-1\right)}^{2}}{\tau }d\Gamma_{t}\geq 0. \end{array}$$
Therefore, Valentini’s entropy $S_{t}$ cannot decrease, and the equality $\frac{d}{dt}S_{t}=0$ occurs iff f=1 corresponding to the quantum equilibrium. This defines an *H*-theorem for BBQT.

## 4. Conclusions and Perspectives

The proposal discussed in this work is certainly schematic but it leads to several interesting conclusions. First, since the dynamics maps used here are deterministic and chaotic, this shows that randomness is unavoidable in BBQT. As stressed by Prigogine [62,63], we have two complementary descriptions: one with trajectories that can be associated with the evolution map $y_{n+1}=F\left(y_{n}\right)$ and the second with a probability density; i.e., as given by the Perron–Frobenius transformation $\tilde{f}\left(y,t_{n+1}\right)={\hat{U}}_{PF}\tilde{f}\left(y,t_{n}\right)$. The two pictures are of course not independent since for a single trajectory we have $\delta \left(y-y_{n+1}\right)={\hat{U}}_{PF}\delta \left(y-y_{n}\right)$ (i.e., $\tilde{f}\left(y,t_{n}\right)=\delta \left(y-y_{n}\right)=f\left(x,t_{n}\right)=\frac{2}{{\left|C\right|}^{2}}\delta \left(x-x_{n}\right)=2L\delta \left(x-x_{n}\right)$). Moreover, for a trajectory, the probability distribution is singular and the convergence to equilibrium is infinitely slow (this is connected to the fact that the coefficients $A_{m}$ in Equation (Equation 51) are given by an integral which is badly defined for the singular Dirac distribution $\tilde{f}\left(y,t_{0}\right)=\delta \left(y-y_{0}\right)$). Therefore, the infinite precision required to compute such a chaotic path (due to the exponentially growing deviation errors with time) leads all practical computations to the strong randomness previously mentioned. To quote Ford [61], “a chaotic orbit is random and incalculable; its information content is both infinite and incompressible”. Subsequently, because of the extreme sensitivity in the initial conditions associated with the predictability horizon and the positive Lyapunov exponent, the use of probability distributions in BBQT seems (at least in our model) unavoidable if we follow Prigogine’s reasoning. Indeed, Prigogine dynamic instability (and thus deterministic chaos) leads to probability. The necessarily finite precision $\delta y_{0}$ used to determine the position of a particle will grow exponentially with time to ultimately cover the whole segment [0,1]. Therefore, if we assign a uniform ignorance probability ${\tilde{f}}_{0}$ over the segment $\delta y_{0}$ (in which the particle is located) then—i.e., subsequently after a few iterations—we will have ${\tilde{f}}_{t}=1$ over the whole segment.

However, we stress that we do not share all the conclusions obtained by Prigogine concerning determinism and probability here (for related and much more detailed criticisms, see e.g., [64]). Indeed, BBQT (as with the classical mechanic considered by Prigogine in [62,63]) is a fully deterministic theory with a clear ontology in the 3D and configuration space. Therefore, while a trajectory could be incalculable by any finite mean or algorithm, the path still fundamentally exists for an idealized Laplacian daemon; i.e., having access to an infinite computing power and precision for locating and defining the particle motion. This metaphor is the core idea of Einstein’s realism: postulating the existence of a world independent of the presence or absence of observers (even if the observers can be part of the world). From this ontic perspective, we need more than simply ignorance in order to justify the use of probability in statistical physics. Indeed, as emphasized long ago by Poincaré, the laws of the kinetic theory of gases still hold true even if we exactly know the positions of all molecules—[65]. There is something objective in the laws of statistical mechanics that goes beyond mere ignorance: otherwise, how could parameters such as diffusion constants have objective physical contents? This point was emphasized by Prigogine from the very beginning, and this constitutes the motivation for his program in order to justify the objectivity of thermodynamics in general and the second law—i.e., irreversibility—in particular.

However, in our opinion, the missing point in Prigogine’s implication—“instability → probability → irreversibility”—is the recognition that in a deterministic theory, the laws (chaotic or not) are not complete but must be supplemented by specific initial conditions, ultimately with a cosmological origin. Indeed, if we suppose a universe made of only one electron described initially by the wave-function $\psi_{0}\left(x,t\right)$ and all the pointers involved in the iterative procedure sketched in Figure 4, then we must use the chaotic Bernoulli map $y_{n+1}=F\left(y_{n}\right)$ for this system or equivalently the Perron–Frobenius evolution $\delta \left(y-y_{n+1}\right)={\hat{U}}_{PF}\delta \left(y-y_{n}\right)$. As we have explained, this system is unstable due to the presence of a positive Lyapunov exponent. Moreover, if we want to make sense of the Formulas (Equation 49) and (Equation 50) with the rapid convergence to $\tilde{f}=f=1$, we must consider a sufficiently regular distribution $\tilde{f}\left(y,t_{0}\right)\neq \delta \left(y-y_{0}\right)$. Now, as mentioned in Section 2, the application of the WLLN to a statistical ensemble requires a “metric” of typicality associated with the Laplacian definition of probability. In BBQT, this metric reads $\rho \left(r,t\right)={f\left(r,t\right)|\psi \left(r,t\right)|}^{2}$, and the law of large numbers leads to Equation (Equation 11)—i.e., $\rho \left(r,t\right)\equiv \lim_{N\to +\infty }\frac{1}{N}\sum_{k=1}^{k=N}\delta^{3}\left(r-x_{k}^{\psi }\left(t\right)\right)$—defined probabilistically in the long term; i.e, for an infinitely long sequence or infinite system. In our problem, this means that we consider an infinite Gibbs ensemble of copies similar to our system, as described in Figure 4. Here, the presence of an infinite sum of Dirac distributions is expected to lead to difficulties in connection with the chaotic map $\delta \left(y-y_{n+1}\right)={\hat{U}}_{PF}\delta \left(y-y_{n}\right)$. In our problem, if the WLLN $\rho \left(r,t\right)\equiv \lim_{N\to +\infty }\frac{1}{N}\sum_{k=1}^{k=N}\delta^{3}\left(r-x_{k}^{\psi }\left(t\right)\right)$ is used to specify the initial distribution at time $t_{0}$, this preserves the chaotic description associated with the positive Lyapunov exponent; therefore, Dirac distributions become problematic. In order to remove this unpleasant feature, one must introduce coarse-graining as proposed by Valentini [32,51]. In our case, this can be done by using a regular weighting function Δ(u) such that $\bar{\rho }\left(x,t\right)=\int du\delta \left(u\right)\rho \left(x-u,t\right)$, which in connection with the WLLN leads to $\bar{\rho }\left(x,t\right)\equiv \lim_{N\to +\infty }\frac{1}{N}\sum_{k=1}^{k=N}\Delta \left(x-x_{k}^{\psi }\left(t\right)\right)$. The coarse-graining of cells in the configuration space plays a central role in the work of Valentini for defining a “subquantum H-theorem” [32,33]. Here, we see that in connection with Prigogine’s work, coarse-graining must be supplemented with a dose of deterministic chaos and entanglement in order to reach the quantum equilibrium regime. We believe that these two pictures complete each other very well.

Before summarizing our work, it is important to go back to Zurek’s envariance as discussed in the introduction in order to see connections with the derivation of Born’s rule as presented in this article. We remind the reader that in 2003, Zurek [66] proposed an alternative proof of Born’s rule based on envariance—a neologism for environment-assisted invariance—with a purely quantum symmetry based on the entanglement of a system with its environment. The importance of this elegant proof could perhaps only be compared with that presented by Gleason [67] in 1957. As stressed by Zurek, “Envariance of entangled quantum states follows from the nonlocality of joint states and from the locality of systems, or, put a bit differently, from the coexistence of perfect knowledge of the whole and complete ignorance of the parts” [66]. The proof is remarkably general and does not rely on any specific ontology, even though it has been used by advocates of the many-world interpretation to justify or recover Born’s rule (for a review and a comparison to the decision-theoretic deduction [5], see [6]).

In order to have a vague idea of the whole derivation, consider a Bell state ${|\Psi \rangle }_{\mathrm{SE}}={|♡\rangle }_{S}{|⟡\rangle }_{E}+{|♠\rangle }_{S}{|♣\rangle }_{E}$ between a system S and environment E. Now, the main idea of envariance concerns symmetry: a local “swapping” (for example, on S for the two possible outcomes ${|♡\rangle }_{S}/{|♠\rangle }_{S}$) in the entanglement is irrelevant for the local physics of E (this is obvious a priori, since E is untouched by the swap). This (unitary) swap reads
(59)$$\begin{array}{c} {|\Psi \rangle }_{\mathrm{SE}}={|♡\rangle }_{S}{|⟡\rangle }_{E}+{|♠\rangle }_{S}{|♣\rangle }_{E}\to {|♠\rangle }_{S}{|⟡\rangle }_{E}+{|♡\rangle }_{S}{|♣\rangle }_{E}={|\Psi^{\prime }\rangle }_{\mathrm{SE}} \end{array}$$
The symmetry of the swap should a priori also impact probabilities associated with outcomes (whatever the definition used for a probability). In other words, if we are allowed to define a probability function $P_{\Psi }{(|♡\rangle }_{S}{|⟡\rangle }_{E})$ for the two correlated outcomes ♡ and ⟡ before the swap, then the previous equation imposes
(60)$$\begin{array}{c} P_{\Psi }{(|♡\rangle }_{S}{|⟡\rangle }_{E})=P_{\Psi^{\prime }}{(|♠\rangle }_{S}{|⟡\rangle }_{E}) \end{array}$$
where $P_{\Psi^{\prime }}{(|♠\rangle }_{S}{|⟡\rangle }_{E})$ is a probability after the swap (i.e., defined for the state $|\Psi^{\prime }\rangle_{\mathrm{SE}}$). Moreover, the swap on S can be compensated by a “counterswap” acting locally on the subsystem E:(61)$$\begin{array}{c} |\Psi^{\prime }\rangle_{\mathrm{SE}}={|♠\rangle }_{S}{|⟡\rangle }_{E}+{|♡\rangle }_{S}{|♣\rangle }_{E}\to {|♠\rangle }_{S}{|♣\rangle }_{E}+{|♡\rangle }_{S}{|⟡\rangle }_{E}={|\Psi \rangle }_{\mathrm{SE}}. \end{array}$$
Now, again from symmetry, we must have the relation
(62)$$\begin{array}{c} P_{\Psi^{\prime }}{(|♠\rangle }_{S}{|⟡\rangle }_{E})=P_{\Psi }{(|♠\rangle }_{S}{|♣\rangle }_{E}). \end{array}$$
However, by comparing Equation (Equation 60) and Equation (Equation 62), we clearly deduce
(63)$$\begin{array}{c} P_{\Psi }{(|♠\rangle }_{S}{|♣\rangle }_{E})=P_{\Psi }{(|♡\rangle }_{S}{|⟡\rangle }_{E})=\frac{1}{2} \end{array}$$
which implies equiprobability for the two branches in the state ${|\Psi \rangle }_{\mathrm{SE}}$. This equiprobablity is clearly an illustration of Born’s rule for the entangled state ${|\Psi \rangle }_{\mathrm{SE}}$. Therefore, envariance can be used to derive Born’s rule (more general reasonings and deductions are given in [66]).

It is important to remark that the reasoning depends on the a priori existence of a probability function, and in order to justify this point, we should rely on a more precise definition of probability in a given ontology. Moreover, in the de Broglie–Bohm ontology, as in classical statistical mechanics, the concept of probability is related to a distribution of particles in ensembles or collectives and is therefore strongly rooted in the concepts of frequency and population. In other words, if we consider a large ensemble of copies for the entangled systems prepared in the quantum state ${|\Psi \rangle }_{\mathrm{SE}}$, then according to the Bernoulli WLLN, the probability $P_{\Psi }{(|♡\rangle }_{S}{|⟡\rangle }_{E})$ is simply a measure of the fraction of systems prepared in the states ${|♡\rangle }_{S}{|⟡\rangle }_{E}$. Now, in the de Broglie–Bohm theory (like in classical physics), *x*-coordinates for particles define a “preferred basis” in the sense that particles are really located at some positions $x_{\Psi }\left(t\right)$ defining trajectories. Zurek’s envariance can thus be applied to the de Broglie–Bohm ontology if we consider systems S and E that are well located in the configuration. Therefore, like in the model used in the present article, we can consider two non-overlapping wave-functions $♡{\left(X_{S}\right)}_{S}$ and $♠{\left(X_{S}\right)}_{S}$ associated with the coordinates $X_{S}$ in the configuration space of the S-subsystem and similarly for the non-overlapping wave-functions $⟡{\left(X_{E}\right)}_{E}$ and $♣{\left(X_{E}\right)}_{E}$ of the E-subsystem. In this ontology, we can give a physical meaning to the invariance under swap or counterswap conditions.

It is indeed possible to postulate that there areas many copies of the systems prepared in the ${|\Psi \rangle }_{\mathrm{SE}}$ state as in the $|\Psi^{\prime }\rangle_{\mathrm{SE}}$ state in the universe. The situation is similar to the one found in a classical gas of molecules were correlated pairs can be defined by exchanging some properties and are present in equal numbers before and after the swap (this kind of symmetry played a key role in the deduction made by Maxwell and Boltzmann justifying the canonical ensemble distribution). Fundamentally, this symmetry in the population is related to some choices in the initial conditions of the whole ensemble. The full deduction of Zurek based on envariance is thus preserved, and this must lead to Born’s rule (at least if we assume that the population of de Broglie–Bohm particles is uniformly distributed in the spatial supports of the various wave functions).

Furthermore, it is important to stress that the envariance deduction is linked to the no-signaling theorem as shown by Barnum [68]. This no-signaling theorem was also emphasized by Valentini [69] in the de Broglie–Bohm theory in order to protect macroscopic causality and to prohibit faster-than-light signaling. Crucially, Born’s rule appears as a necessary condition for the validity of the no-signaling theorem (this was also related to the second law of thermodynamics by Elitzur [70]). Interestingly, in the present work, we considered regimes of quantum-nonequilibrium where the symmetry of the entangled wave-functions was not present in the particle distribution characterized by the $f(X_{t},t)$ function. However, in the end, we showed that if the environment of pointers was already in quantum equilibrium, then the system would be driven to the quantum equilibrium f=1 acting as an attractor under the chaotic Bohmian dynamics. In the end, this also shows that the quantum equilibrium in the de Broglie–Bohm dynamics is natural and also how fragile and unstable physical deviations to the Born rule are. We believe that this confirms the deductions made by Zurek concerning the fundamental role of envariance.

There is another way to express the same concept: going back to our discussion about typicality at the end of Section 2, we see that in this article, we indeed developed a model that does not assume quantum equilibrium for all particles. The system moving in the interferometer is initially out of quantum equilibrium f=1. However, it is quickly driven to quantum equilibrium due to (1) entanglement with pointers already relaxed in the regime f=1 and (2) the presence of chaotic dynamics inducing fast mixing and thus a fast relaxation f→1. It is interesting that the number of iterations *N* and therefore the number of pointers involved in the process does not have to be large (i.e., we do not have to go to the thermodynamic limit N→+∞ associated with a quantum bath). As we have shown, the chaotic Bernoulli map drives the system to quantum equilibrium already for N≃3. This demonstrates, we think, the robustness of this attractor leading to Born’s rule.

To summarize, in this work, we have proposed a mechanism for relaxation to quantum equilibrium in order to recover Born’s rule in BBQT. The proposed mechanism relies on entanglement with an environment of “Bohmian pointers” allowing the system to mix. The scenario was developed for the case of a single particle in 1D motion interacting with beam splitters and mirrors, but the model could be generalized to several situations involving collisions between quanta and scattering with defects or other particles. The general proposal is thus to consider the quantum relaxation to Born’s rule as a genuine process in phases of matter where interactions between particles play a fundamental role. This involves usual condensed matter or even plasma or gases where collisions are mandatory. For example, based on our toy model, we consider that interaction with the beam splitter and entanglement with Bohmian pointers is a good qualitative model for discussing collisions between molecules in the atmosphere, and if we remember that nitrogen molecules at a temperature of 293 K and at a pressure of 1 bar involve typically a collision frequency of 7 × $10^{9}$ /s (which implies fast dynamics for reaching quantum equilibrium), we thus have a huge number *n* of collisions per second corresponding to a huge number of iterations in our Bernoulli-like process based on the Perron–Frobenius operator $f\left(y,t_{n+1}\right)={\hat{U}}_{PF}f\left(y,t_{n}\right)$. Compared to Valentini’s framework [32,33] where mixing and relaxation to quantum equilibrium are associated with coarse-graining à la Gibbs, our approach emphasizes the role of information losses due to entanglement with a local environment. In both cases, we obtain an increase of entropy and a formulation of the H–theorem for BBQT. These two views are certainly complementary, in the same way that Gibbs and Boltzmann perspectives on entropy are related. This could have an impact on the efficiency of quantum relaxation in the early stages of the evolution of the universe [37,38].

## Figures and Tables

**Figure 1 entropy-23-01371-f001:**
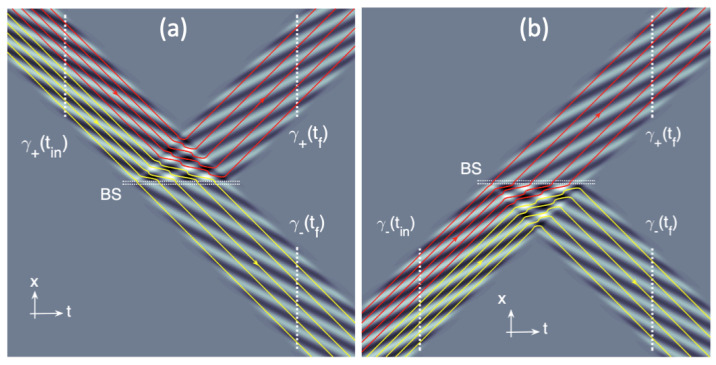
(**a**) Scattering of a 1D wave-packet impinging on a 50/50 beam-splitter (BS). The colormap shows Re[Ψ(x,t)] in the t,x plane. The color (red and yellow) lines are de Broglie–Bohm trajectories associated with this wave-function (red and yellow trajectories are ending in two different wave-packets. The dotted white lines are crosscuts, as discussed in the main text. (**b**) A similar situation when a wave-packet impinges on the other input gate.

**Figure 2 entropy-23-01371-f002:**
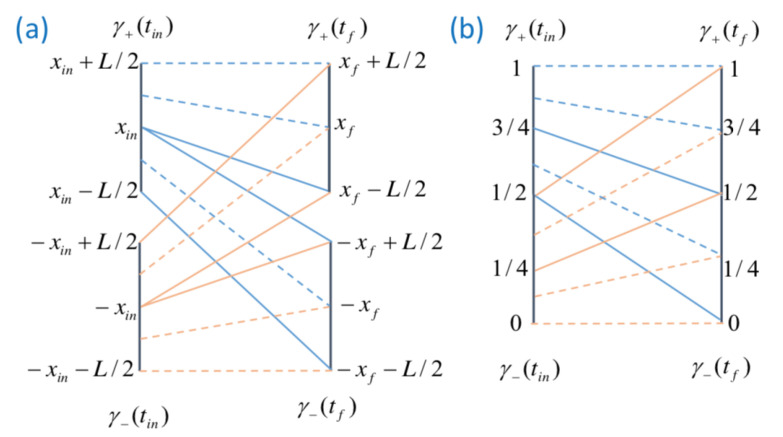
(**a**) Transformation from the initial $\gamma_{\pm }\left(t_{in}\right)$*x*-space to the final $\gamma_{\pm }\left(t_{ou}\right)$*x*-space for the two situations shown in Figure 1a,b, respectively (depicted as blue lines and red lines, respectively). (**b**) The same transformation using the *y* coordinate instead of the *x* coordinate (as explained in the main text).

**Figure 3 entropy-23-01371-f003:**
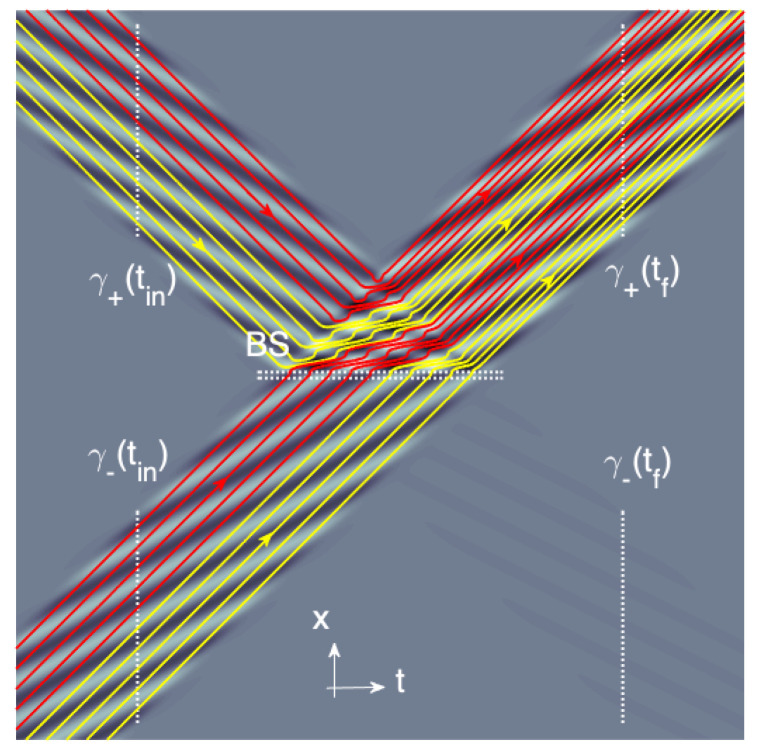
Same as in Figure 1 but for a symmetric superposition of the two wave-functions impinging from the two input gates of BS. The superposition principle forces the resulting wave-packet to end its journey in the exit gate $\gamma_{+}\left(t_{f}\right)$. The pilot-wave dynamics are strongly impacted by the linearity of the superposition (compare with Figure 1).

**Figure 4 entropy-23-01371-f004:**
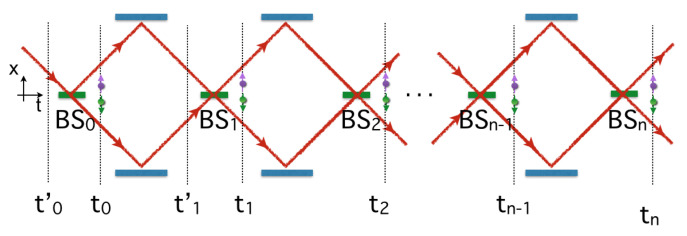
Drawing of the iterative procedure for entangling an initial wave-packet with “Bohmian” pointers providing unambiguous which-path information on the pilot-wave particle motion (as explained in the main text). The various pointers interacting at time $t_{0}$, $t_{1}$ …are sketched as qubit states.

**Figure 5 entropy-23-01371-f005:**
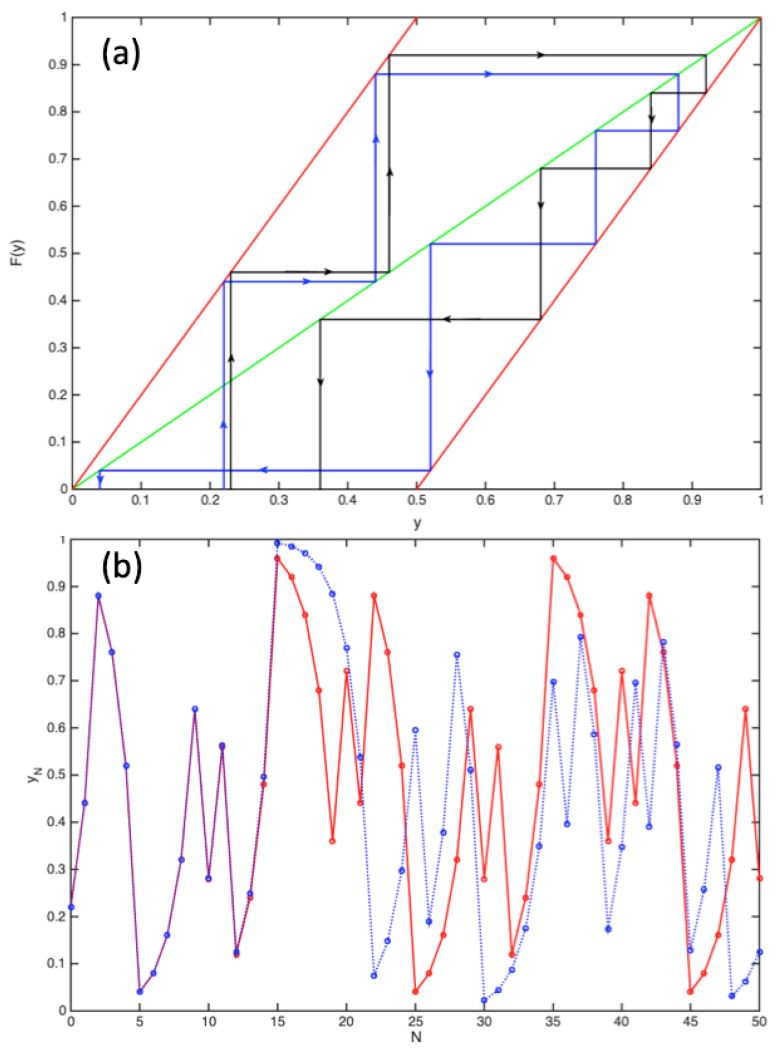
(**a**) Bernoulli map $y_{n}=F\left(y_{n-1}\right)$ in the $y,y^{\prime }$ plane where the function $y^{\prime }=F\left(y\right)$ acts iteratively. The red and green lines are acting as mirrors during the process. The black and blue trajectories correspond to different initial coordinates $y_{0}=0.22$ and $y_{0}=0.23$. (**b**) The same Bernoulli map is shown as a function y=y(n) of the iteration steps n=0,1,⋯. The two chaotic trajectories shown in red and blue correspond to $y_{0}=0.22$ and $y_{0}=0.220001$, respectively (see main text).

**Figure 6 entropy-23-01371-f006:**
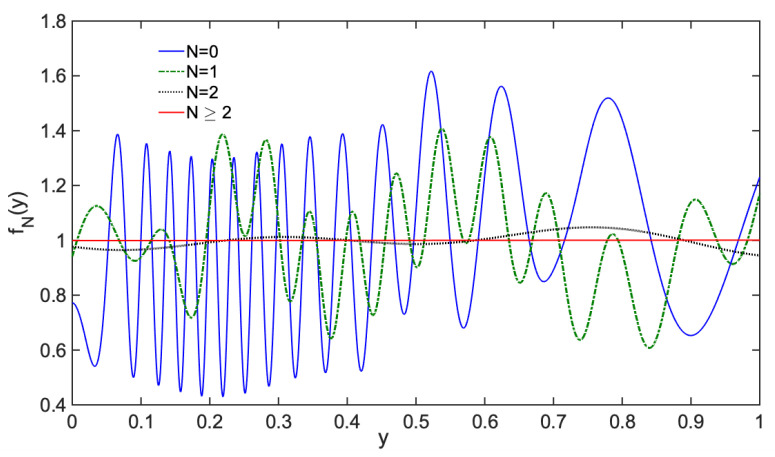
Evolution of $\tilde{f}\left(y,t_{n}\right):={\tilde{f}}_{n}\left(y\right)$ as a function of *y* for a few *n* values (using the Perron–Frobenius operator Equation (Equation 49)). The initial distribution ${\tilde{f}}_{0}\left(y\right)$ (blue curve) was chosen to be arbitrarily irregular. After a few iterations n≥2, the function ${\tilde{f}}_{n}\left(y\right)$ cannot be distinguished from the line $\tilde{f}=f=1$ associated with quantum equilibrium (i.e., Born’s rule).

## Data Availability

Not applicable.

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
