# Peer review of "Justifying Born’s Rule Pα = |Ψα|2 Using Deterministic Chaos, Decoherence, and the de Broglie–Bohm Quantum Theory"

_entropy, 2021, doi:10.3390/e23111371_

Round 1

Reviewer 1 Report

The Entropy manuscript "Justifying Born's rule ... using deterministic chaos and the de Broglie - Bohm quantum theory" develops an interesting toy model in which a single-particle system (basically a one-dimensional particle evolving in a kind of repeated interferometer) interacts intermittently with outside "pointer" degrees of freedom.  It is shown that the system particle trajectory exhibits a kind of deterministic chaos and that the system probability distribution evolves inevitably (in an appropriate sense) toward the born rule or quantum equilibrium distribution.  

This is a strong piece of work which is certainly suitable for publication in the special issue honoring the work of Zurek.  My only real reservation about the paper is that at times the author perhaps somewhat overstates or misrepresents certain foundational points in the ongoing debates (to which this paper certainly contributes something) about how to understand the Born rule in the context of the dBB / pilot-wave theory.  For example, the author seems slightly dismissive of the "typicality" account of born rule statistics (championed especially by Duerr, Goldstein, and Zanghi) on the grounds that that view is at least somewhat circular.  I don't think that view is actually circular, though I appreciate that it may appear that way to some.  But however one feels about that, it is certainly not the case that the "dynamical relaxation" approach (championed especially by Valentini and developed in the present paper) is free from any similar appearance of circularity.  For example, in what may appear to be only an innocuous side remark, the author here notes his assumption that "the pointers are all in quantum equilibrium".  But this raises the possible appearance that the system perhaps inherits its quantum equilibrium from the pointers, for which this property has been assumed rather than explained.  Relatedly, the author perhaps over-states what has been shown when he writes that "Valentini's entropy S_t cannot decrease".  Just as in classical statistical mechanics, it is true and will always be true that there exist possible initial states for which the long-term behavior is "weird" in the sense of violating the sort of "relaxation to equilibrium" behavior that is discussed here.  Any analysis which purports to prove that relaxation to equilibrium is in some absolute sense inevitable is simply wrong.  The most that might be true is that relaxation to equilibrium is typical, i.e., will occur for virtually (but not precisely!) all possible initial conditions.  But acknowledging this fact forces one to acknowledge that even in attempting to avoid the supposed appearance of circularity in the direct argument that Born rule statistics are typical and hence in need of no further explanation (e.g., via dynamical relaxation), one still cannot completely avoid making arguments about what is and isn't typical, which of course requires the use of some measure of typicality on the set of possible initial states, which is the very thing that supposedly makes the direct typicality arguments appear circular. 

So, OK, maybe a slightly more even-handed discussion of these background foundational questions would be appropriate.  It might even increase the readership and influence of the paper.  But a little rhetorical excess can be forgiven in a paper, like this, of generally high quality, so if the author wants to stand by his way of contextualizing the new work, I wouldn't ultimately have any problem with it and certainly wouldn't want that to get in the way of the paper's publication. 

Reviewer 2 Report

See my comments in the attached file, REPORT.docx
